# Highly reinforced and degradable lignocellulose biocomposites by polymerization of new polyester oligomers

Erfan Oliaei [1,2], Peter Olsén [2] ✉, Tom Lindström[1] & Lars A. Berglund [2] ✉

Unbleached wood fibers and nanofibers are environmentally friendly bio-based candidates for material production, in particular, as reinforcements in polymer matrix biocomposites due to their low density and potential as carbon sink during the materials production phase. However, producing high reinforcement content biocomposites with degradable or chemically recyclable matrices is troublesome. Here, we address this issue with a new concept for facile and scalable in-situ polymerization of polyester matrices based on functionally balanced oligomers in pre-formed lignocellulosic networks. The idea enabled us to create high reinforcement biocomposites with well-dispersed mechanically undamaged fibers or nanocellulose. These degradable biocomposites have much higher mechanical properties than analogs in the literature. Reinforcement geometry (fibers at 30 μm or fibrils at 10–1000 nm diameter) influenced the polymerization and degradation of the polyester matrix. Overall, this work opens up new pathways toward environmentally benign materials in the context of a circular bioeconomy.

Cellulose biocomposites from nanocellulose or plant fibers, with polymer matrix, often do not sufficiently contribute to sustainable development, are not biodegradable and the processing approach is not scalable[1]. In addition, mechanical properties of cellulose biocomposites are often insufficient to replace established materials, because of low fiber content. In a circular bioeconomy, constituents should be recovered, in addition to green synthesis routes and bio-based building blocks[2–4]. This work tries to address both the challenge of circularity and low fiber content by creating new, green, and hydrolytically degradable oligomers for in-situ polymerization within high-content lignocellulose reinforcement networks.

Fossil-based plastics are often incinerated or disposed of as landfill[5]. Plastic waste is long-lasting in nature, physically harming wildlife and providing chemical hazards to the environment[6]. Plastic production and plastic waste incineration result in 400 Mt of $CO_2$ emissions per year[7]. Conventional plastics contributed 1.7 Gt $CO_2$-eq emissions in 2015 throughout their life cycle, which will increase to 6.5 Gt $CO_2$-eq by 2050 if the current increase in use continues[8]. However, by combining measures for reduced energy, materials aspects, recycling, demand-management, and substitution of fossil-based feedstock with biomass, these levels could be reduced[8]. Circular economy concepts emphasize reduced fossil-based plastic manufacturing, reuse, and recycling[9]. Composites based on lignocellulosic plant fibers provide $CO_2$ storage, since plant tissue synthesis uses solar energy and sequesters $CO_2$ to create plant cell wall polymers[2,4,10,11]. One challenge with biocomposites with short service life is the difficulty of extracting polymer and fiber constituents for recycling at end-of-life[5]. Mechanical recycling is vital to extend the material life cycle[5,12,13], but chemical recycling is of great interest as an end-of-life scenario for more "sustainable" biocomposites[7,14,15].

The dominant industrial process for polymer matrix biocomposites, melt-compounding, is limited in terms of fiber content, results in mechanical fiber damage, and has issues with reinforcement aggregation, particularly for nanocellulose reinforcements[16].

[1]RISE Bioeconomy and health, Stockholm, Sweden. [2]Wallenberg Wood Science Center, Department of Fibre and Polymer Technology, KTH Royal Institute of Technology, Stockholm, Sweden. ✉e-mail: polsen@kth.se; blund@kth.se

Melt-processing is energy-intensive[3] as melt-processed biocomposites are typically heated and cooled multiple times to produce the final product. Most molded plant fiber biocomposites in use today are based on melt-processing of thermoplastic-plant fiber mixtures[17] and are not degradable. Processing by in-situ polymerization has lower cumulative energy as the polymer is formed during biocomposite manufacture. Biobased epoxies are commonly used for load-bearing biocomposites[18–20] but they are often not chemically recyclable. Aliphatic polyesters are interesting in this context as they are degradable to non-hazardous products in industrially relevant environments[21,22]. A challenge is that most semi-crystalline aliphatic polyesters have low glass transition temperature ($T_g$), which translates into low modulus and tensile strength[23].

A new biocomposite concept would be to combine stiff plant fiber or nanocellulose networks with a comparably low content of hydrolyzable polymer matrix suitable for chemical recycling[10,24–26]. In order to reach high fiber content, in-situ polymerization within a pre-formed fiber network is a feasible approach. In-situ polymerization of non-degradable polymers in cellulose fiber networks has been carried out through free radical polymerization[27,28], controlled radical polymerization[29], and ring-opening metathesis polymerization[30]. There is, however, little work using degradable polymers for high fiber content composites. The reason is that degradable polymer matrix synthesis is often based on ring-opening polymerization (ROP) of cyclic monomers[31–35]. The synthetic challenge is that cellulosic fibers contain more than 5 wt% moisture under ambient conditions. Water molecules will initiate polymerization which results in low molar mass polymers (clarified in Supplementary Discussion 1). For in-situ ROP, this problem is aggravated at high lignocellulose content since total moisture content is increased. Moisture also hinders polycondensation/polyesterification reactions of linear monomers towards high molecular weight polymers. Thus, polymerization must be performed under extremely dry conditions (distilled under inert gas flow, high vacuum, and/or very high temperatures[34,36–41]) or in organic solvents[42]. This impedes the sustainability of the process, as well as its applicability in scalable composite production.

Other strategies are exemplified by the linking of different degradable oligomers to high molecular weight thermoplastics and crosslinked thermosets[43]. The chemistry used includes direct use of isocyanates[44,45], acrylate addition followed by either free radical polymerization[46,47] or thiol-ene chemistry[48], reversible Diels-Adler networks[49], and silane chemistry[50]. For sustainable development, the in-situ polymerization system ideally should be insensitive to moisture, retain degradability after polymerization and preferably show interface reactivity for interfacial adhesion purposes and rely on nontoxic and benign chemicals. Here, a stoichiometrically balanced, novel three-arm oligomer based on ε-caprolactone is a model precursor for the polymerization system. After infiltration, oligomers polymerize thermally via esterification conjugation to high molecular weight polymers, within well-dispersed wood fiber or microfibrillated lignocellulose (MFLC) reinforcement networks. This approach removes the need for tedious drying and toxic reactants, and the polymerization system is applicable at mild conditions (atmospheric pressure and low temperature curing with no lignocellulose degradation risk). The system was applied to obtain high volume fraction reinforcement. The approach paves the way for scalable production of advanced and degradable biocomposites with much improved properties. In addition, wood fiber and nano-lignocellulose reinforcements are compared with respect to mechanical behavior and degradation.

Here we are investigating wood-based reinforcements, both fibers and nanocellulose microfibrils since they can form strong biocomposites[51–53]. Mechanical property effects and end-of-life characteristics for wood fiber and lignocellulosic microfibril network reinforcements will be compared. Lignocellulosic wood fibers and microfibrils are of particular interest for composites for sustainable

development. The investigated biobased unbleached kraft fibers show higher yield, lower energy demand, carbon footprint, and are subjected to less severe chemical treatment than cellulosic wood pulp fibers of low lignin content[54,55]. Important eco-indicators of high-lignin unbleached fibers such as cumulative energy demand (7–11 MJ/Kg), global warming potential (-0.2 Kg $CO_2$ eq/Kg), and water depletion (0.02–0.04 $m^3$/Kg) values show relatively low environmental impact and are typically 30% lower than bleached fibers[55]. Furthermore, they show little mechanical damage and favorable compatibility with non-polar polymers. MFLC investigated here is also characterized by high yield, reduced water sensitivity, and better compatibility with "hydrophobic" polymers compared to higher-purity microfibrillated cellulose[54–56], although they are not showing interesting cumulative energy demand (>100 MJ/Kg)[54]. Note that MFLC is used for mechanical properties comparison with fibers, not environmental aspects. In this study, we developed an in-situ polymerization system and showed the greenness of this chemical process and its low environmental impact using key green chemistry metrics (atom economy, reaction mass efficiency, and E factor).

The present approach allows for high reinforcement content, good reinforcement dispersion, and no mechanical damage to the reinforcement. The in-situ polymerization process should also improve lignocellulose/polymer interfacial adhesion[42,57]. PCL is investigated as a model polymer representing soft, low $T_g$ semicrystalline and compostable aliphatic polyester as candidate replacement for polyethylene (PE) and polypropylene (PP) biocomposite matrices. Although neat, unmodified PCL shows low mechanical properties, the present wood fiber biocomposites with crosslinked, caprolactone-based c-PCL matrix show better mechanical properties than previous reports for biocomposites with aliphatic polyester matrix[58–62], since the processing concept allows high content of well-dispersed, efficient network reinforcements, Fig. 1. Chemical durability, shaping capability, and hydrophobicity of c-PCl biocomposites, on the other hand, are not competing with polyolefins. Overall, the present approach based on novel aliphatic polyester oligomers and reactive processing, suggests the possibility to replace non-degradable plastics and composites with degradable biocomposites.

## Results

### Caprolactone oligomers and polymerization

The feasibility of reactive processing by in-situ polymerization of aliphatic polyesters in lignocellulose reinforcements is high, but is

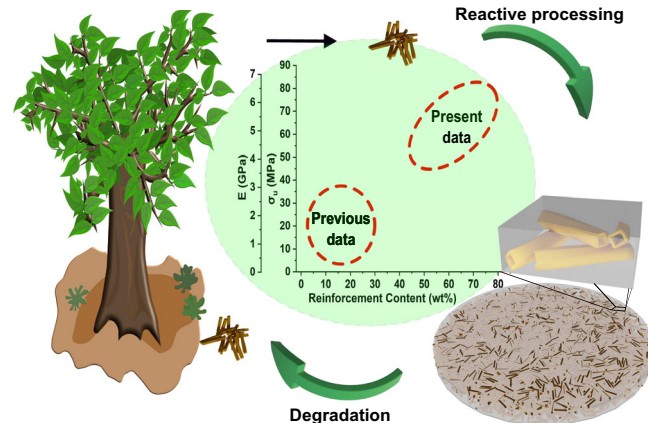

**Fig. 1 | A schematic representation of the concept presented in this work.** It highlights the circular economy principle by creating materials from biobased resources (wood fibers) employing reactive processing (in-situ polymerization and densification), which could then degrade at the product's end of life. This method has the potential to produce biocomposites that are much stronger than typical biocomposites due to the high content of well-dispersed reinforcement.

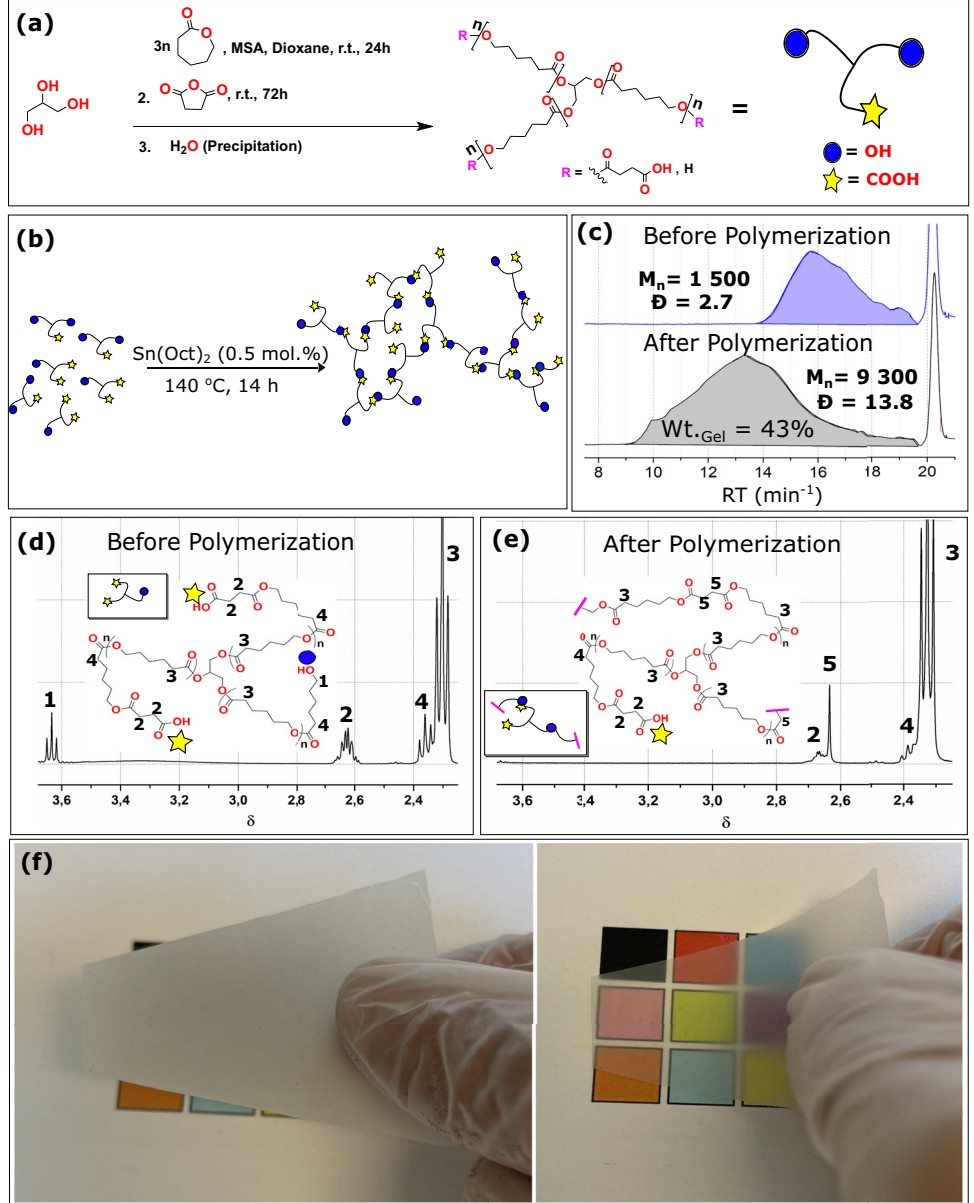

**Fig. 2 | Insights into the procedures of the polyester oligomer formation and polymerization into polymeric film. a** Synthesis of functional caprolactone oligomers. **b** Schematic representation of the oligomer polymerization system catalyzed by Sn(Oct)$_2$. **c** Number average molecular weight (M$_n$) and dispersity index (Đ) of the oligomers before and after polyesterification, the measurement is performed on the extracted, soluble polymer fraction. **d** $^1$H NMR on the starting functional three-arm copolymer, the pink part indicates the continuation of the network structure. **e** $^1$H NMR on the extracted polymer after polymerization. **f** Images of homopolymer sheets after polycondensation. Optical transmittance is fairly high (right image), with considerable light scattering and haze (left image). Film thickness ≈135 μm.

commonly hindered by moisture interference with polymerization. Here, new oligomers ("prepolymers") are synthesized and polymerized (cured) by polyesterification alone (Fig. 2a) or in the presence of a high content (>20 wt%) of moisture-containing reinforcement (Fig. 3a). Biocomposites processing takes place in two steps: in-situ polymerization followed by hot-press consolidation, which could include shaping, Fig. 3a and Supplementary Fig. 32b. Three functional oligomers of different molecular weights and COOH:OH end group ratios were synthesized. The oligomer preparation method is sequential ring-opening polymerization of ε-caprolactone (εCL), initiated by glycerol (Gly), and catalyzed by methanesulfonic acid (MSA), followed by end-capping with succinic anhydride (SA) which was stopped at 50% end-group conversion, Fig. 2a and Supplementary Table 1. MSA is employed as a catalyst because it is highly active for the polymerization of CL, allowing for

polymerization at ambient temperature. Acid-catalyzed ring-opening of SA is slower than for εCL, so that the degree of transesterification is diminished while simultaneously enabling high control over the end-group conversion (to R-COOH) to produce functionally balanced oligomers.

In the present system, a near-complete conversion of εCL was achieved for all three oligomers after 24 h. The end-capping reaction was monitored by $^1$H NMR; after 72 h, 50% end-group conversion of hydroxyl to carboxylic acid was achieved, Supplementary Table 1 entries [2] and [3]. The yield depended on the theoretical molecular weight of oligomers, where a larger feed ratio resulted in a higher yield due to the better precipitation, Supplementary Table 1 entry [1] to [3]. The final multifunctional oligomers are three-armed aliphatic ester oligomers with two types of functional groups: carboxylic acids and hydroxyls (stoichiometrically balanced), Fig. 2d.

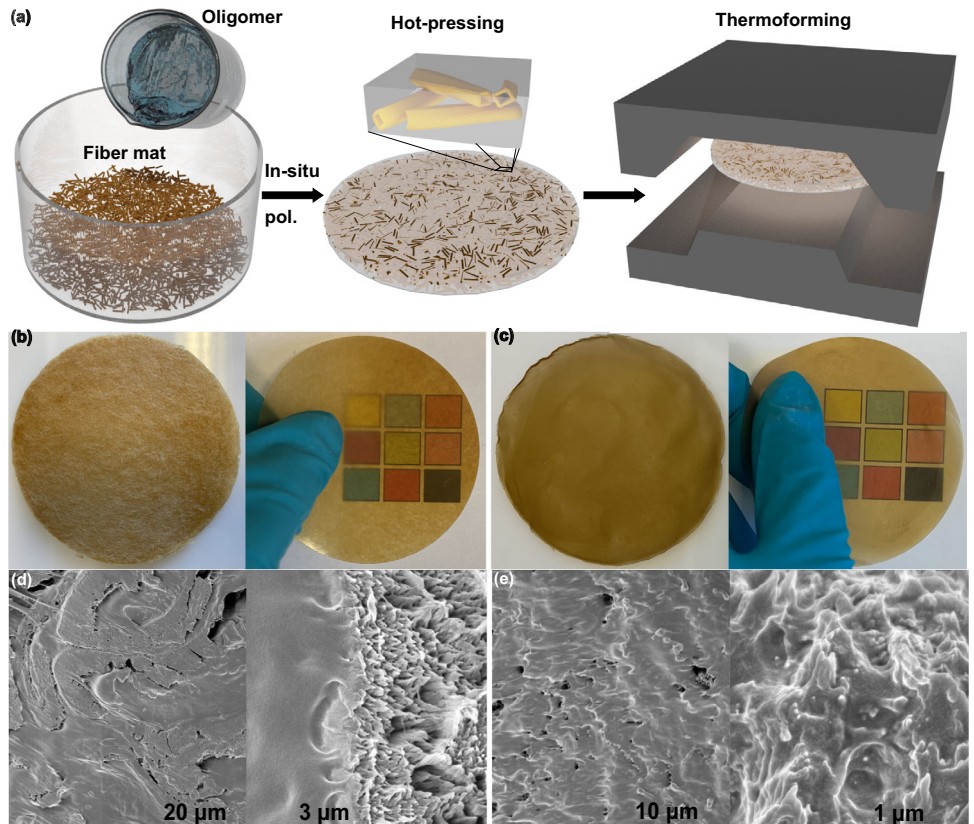

**Fig. 3 | Schematically depicted preparation procedures of the biocomposites as well as their macroscopic and microscopic structures. a** Processing steps: impregnation of fiber mat with oligomer liquid, in-situ polymerization, followed by hot-pressing. The wood fiber (WF) polymer matrix biocomposite is then subjected to thermoforming for consolidation. In industrial process, thermoforming would also be a shaping step. **b** Photographs of 25% WF biocomposites from wood fibers before and after hot-pressing; sample diameter = 7.2 cm. **c** Photographs of nanostructured 27% microfibrillated lignocellulose (MFLC) biocomposites before and after hot-pressing; sample diameter = 7.2 cm. **d** Cross-sectional SEM images from cryo-fractured, hot-pressed 25% WF biocomposite. **e** Cross-sectional SEM images of cryo-fractured, hot-pressed 27% MFLC biocomposite.

Lignocellulose reinforcements typically contain moisture at ambient conditions, ~5–10% at room temperature and 50% relative humidity, which limits the selection of a catalytic system for in-situ polymerization of polyesters. The considered polymerization reaction is direct esterification via the combination of carboxylic acids and hydroxyls with the release of water. Commonly, acid catalysis by e.g. toluene sulfonic acid is used[63]. This strategy, however, is not applicable for lignocellulose reinforcements because of fiber degradation effects. Initial polymerization trials were performed with neat oligomers under uncatalyzed conditions. The degree of polymerization was then insufficient, see number average molecular weight, $M_n$, and gel content, Supplementary Table 2 entries [1] to [3]. For this reason, Sn(Oct)$_2$ was used as catalyst (1 mol%) and CL-oligomer polymerization was performed at 140 °C. Under these conditions, the molecular weight and gel fraction became too high for the hot-pressing stage, Supplementary Table 2 entries [4] to [6]. The amount of Sn(Oct)$_2$ was then reduced to 0.5 mol%, which resulted in an insoluble gel content of 43 % while $M_n$ was still high enough, Supplementary Table 2 entry [7]. These results are optimal in that liquid flow takes place under hot-pressing so that consolidation (reduced porosity) becomes straightforward, as discussed in the next section.

The polymerization proceeded via esterification between COOH and free OH end-groups on neighboring oligomers, Fig. 2d, e. The $^1$H NMR spectra before and after polymerization gave insights into the in-situ polymerization mechanism. After polymerization, a partially crosslinked PCL (c-PCL) polymer as a solid sheet of material is formed, Fig. 2f. In the hot-pressing stage, the state of c-PCL allows for liquid flow and plastic deformation, mechanisms that facilitate consolidation, densification, and shaping.

## In-situ polymerization of CL-oligomers in lignocellulose reinforcement

The wood fibers in this study are 39 µm in diameter and 2.5 mm in length on average[64], whereas the present MFLC is 2.5–1000 nm in diameter and 5–50 µm in length[56]. The initial porosity of the network reinforcements was ~85%, although polymerization followed by hot-pressing densified the system. For in-situ polymerization, CL oligomers and the catalyst were added to the system and impregnated into the networks (wood fibers or MFLC with the same compositions and sources), to form model biocomposites with favorable polymer matrix distribution. The reaction was carried out at 140 °C for 14 h and then samples were hot-pressed at 40–50 MPa and 120 °C for 5 min (Fig. 3a).

The selected polymer system is based on the oligomer GLY-PCL$_{14}$-SA$_{1.5}$ ($M_n$ 1500) and catalyzed with 0.5 mol% of Sn(Oct)$_2$, Supplementary Table 2 entry [7]. After polymerization, it has a 43% gel fraction (sparsely crosslinked with low $T_g$) and a soluble PCL fraction with $M_n$ 9300. This system enables both some further polymerization and polymer flow during secondary hot-pressing for consolidation (Fig. 3). This oligomer system was therefore selected for polymerization in all biocomposite formulations. Since the gel content is moderate after in-situ polymerization, the biocomposites can be post-processed, e.g. by shaping into complex 3D structures (Supplementary Fig. 32b).

The very large size difference between wood fibers (WF) and MFLC is apparent in Supplementary Fig. 26a, b. Wood fibers (spruce) are typically 30 µm by 3 mm[55]. The width of 44 wt% of fibrils is <100 nm,

and the width of 56 wt% is between 100 nm and 1 μm. Therefore, the main differences are related to size, specific surface area and intrinsic mechanical properties. Cryo-fractured cross-sections of hot-pressed MFLC and WF biocomposites with c-PCL in Fig. 3d, e illustrate favorable MFLC dispersion at sub-micrometer scale and show WF/c-PCL microstructure at 50-μm scale. Some biocomposites' porosity is apparent, up to 10%, Supplementary Table 1. The plastic flow characteristics of the matrix are apparent from the smeared appearance in the images (Fig. 3d, e and Supplementary Fig. 27). The fiber-polymer interfaces do not show apparent signs of debonding, suggesting good interfacial wood fiber/c-PCL adhesion, probably related to chemical fiber-polymer reactions (Fig. 3d and Supplementary Fig. 27). Figure 3d also shows c-PCL is also distributed in the wood fiber lumen region (inside the cell wall), also confirmed by Supplementary Fig. 28. Micro- and nanostructural differences between WF and MFLC biocomposites influence the reinforcement mechanisms. The larger pore size of WF biocomposites (Fig. 3d, e) and higher porosity (Supplementary Table 1) confirm that c-PCL matrix distribution depends strongly on the reinforcement (WF or MFLC). Biocomposite surfaces are flat and rather smooth with no distinct pattern from the reinforcement network, confirming plastic flow during hot-pressing, Supplementary Fig. 29.

In Supplementary Fig. 22, it is apparent that thermomechanical biocomposite properties depend on the reinforcement type. This may not only be caused by reinforcement efficiency aspects, but also by polymerization effects on polymer structure and properties. A detailed analysis of polymer formation effects was therefore performed on biocomposites from the two reinforcement types (WF and MFLC nanofibers), namely PCL/MFLC41 and PCL/WF44 with about 40 wt% reinforcement (Supplementary Table 3).

Careful end-group analysis (of dissolved c-PCL) revealed that the degree of oligomer polyesterification decreased in the presence of the reinforcement, see conversion end-groups for c-PCL in MFLC41 and WF44, Supplementary Table 3. For c-PCL/WF44, the acid end-group conversion from neat c-PCL was reduced by 12% (53% vs 65%), whereas it was reduced by as much as 27% (38% vs 65%) for the higher specific surface c-PCL/MFLC41. Oligomer acid end-groups are likely reacting with surface hydroxyls on the WF or MFLC reinforcements. The gel content was defined as the fraction of c-PCL that could not be dissolved (see Characterization section). Shorter polymer chains are readily soluble, whereas higher molar mass molecules have lower solubility. Furthermore, polymer chains strongly linked to the reinforcement cannot be extracted. The gel content of the polymer is similar for neat c-PCL (43%) and c-PCL/MFLC41 biocomposite (46%). In contrast, c-PCL/WF44 showed a gel content as high as 75% since oligomer crosslinking was favored. Polymer trapped or bonded inside the wood fiber cell wall or lumen may contribute to the higher gel content of c-PCL/WF44 compared to c-PCL/MFLC41. The hypothesis for the lower c-PCL gel content in MFLC biocomposites is that short c-PCL oligomers are reacting with the surface hydroxyls of MFLC nanofibers. The specific surface area of the MFLC reinforcement is much higher than for WF, which could influence polymerization. If we assume an average cylindrical nanofiber diameter of 20 nm and organize all nanofibers in the same direction with square packing, the average distance between nanofibers will be only ~10 nm at a nanofiber volume fraction of 0.35[65]. Consequently, more active sites on the MFLC reinforcement can participate in the reaction, and the stoichiometric balance of functional groups in oligomers (COOH:OH 50%:50%) is disturbed during condensation reactions. The higher accessible surface area of MFLC increases the extent of reactions in the interface region, resulting in a greater number of short c-PCL oligomers bonded to the MFLC fibrils. This limits the overall c-PCL network formation by changing the reactant stoichiometry. To test the hypothesis, the reaction time was reduced to 3 h for both MFLC and WF reinforcements, and free polymer chains were removed from the system by excessive solvent washing. After 3 h of reaction and polymer

extraction, the MFLC system had a much higher carbonyl intensity from polyesters than the WF system. It means that the MFLC reactions with oligomers (covalent oligomer attachment to MFLC, and formation of ester linkages) are much more frequent, in support of the hypothesis, Supplementary Fig. 23.

The swelling of biocomposites in organic solvents was also investigated. The gel fraction in the neat c-PCL polymer is insoluble but shows high swelling in organic solvents (Supplementary Fig. 24). The reason for "high" swelling is that the crosslink density is low and the solubility of PCL is high in the solvents. The c-PCL/MFLC41 and c-PCL/WF44 biocomposites show much lower swelling than neat c-PCL due to swelling restrictions from the reinforcement (interfiber bonding). MFLC composites showed the lowest swelling because of "stronger" and more numerous interactions between MFLC reinforcement nanofibers.

## Crystallization and thermal properties of biocomposites

Since crosslinking is sparse, the polymer matrix is semi-crystalline. Dried biocomposites were heated to 120 °C and then cooled to −80 °C. Afterward, the samples were heated from −80 °C to 150 °C. DSC curves from heating and cooling of biocomposites of different reinforcement content are represented in Supplementary Fig. 22, so that thermal history is removed and crystallization conditions are the same. Pure c-PCL showed two distinct melting peaks related to two different populations of crystal size. C-PCL biocomposites, however, show one distinct melting peak. The reason may be due to a nucleation effect from the lignocellulose reinforcement. $T_m$ of biocomposites shifts to higher temperatures with increasing reinforcement content due to interactions[66] between c-PCL matrix chains and reinforcement, Supplementary Table 5. Grafted c-PCL chains from the reinforcement may influence the matrix c-PCL behavior. $T_c$ of the composites shifts to higher temperatures with increased reinforcement content, in support of nucleation effects from fibers.

When thermal history is removed, c-PCL in all biocomposites except for c-PCL/77% HP-WF shows a lower degree of crystallinity than pure c-PCL, Supplementary Fig. 22b, c and Supplementary Table 5. The reason is not completely clear, but gel content is certainly one reason, since it is higher for biocomposites. The lower crystallinity for MFLC composites is due to the population of grafted PCL chains on MFLC and/or nanoconfinement effects, which limits crystallization. Note that the degree of crystallinity reported in Table 1 is related to the original state of the biocomposites (prepared similarly), without thermal history removal. These c-PCL matrix data are for biocomposites used in mechanical testing and will influence mechanical properties of biocomposites. In Fig. 4c, degree of crystallinity in c-PCL matrix versus gel content shows interesting trends. Reduced crystallinity with gel content is because crosslinked c-PCL domains show constrained molecular mobility, preventing crystallization. c-PCL in biocomposites has a higher gel content and lower crystallinity than pure c-PCL. One reason is that oligomers react with the carboxyl and hydroxyl groups of the reinforcement surfaces, increasing the fraction of gel content. There is, however, not a clear monotonous trend of rising gel content (or reduced crystallinity) with lignocellulose content due to the complexity of the system. In wood fiber biocomposites, increasing fiber content generally results in higher gel content. One reason may be c-PCL trapped or bonded inside wood fibers or in lumen region. The high crystallinity of the hot-pressed 77 wt% WF biocomposite correlates with low gel content and possibly a more favorable thermal history for crystallization.

## Mechanical properties

Figure 4 demonstrates strong improvement in c-PCL properties with lignocellulose reinforcement and also shows better properties than for comparable PCL biocomposites reported in the literature. In Fig. 4a, stress-strain curves are presented for c-PCL/WF and c-PCL/MFLC

**Table 1 | Summary of physical and mechanical properties of WF and MFLC biocomposites**

| Sample | Thickness (µm) | Apparent density (g/cm³) | Apparent porosity (%) | Polymer gel content (wt%) | Polymer degree of crystallinity (%) | Elastic modulus (GPa) | Ultimate tensile strength (MPa) | Strain to failure (%) |
|---|---|---|---|---|---|---|---|---|
| c-PCL | 135.3 (12.6) | 1.18 (0.16) | 0 | 43 (2) | 44.0 | 0.5 (0.1) | 11.5 (1.2) | 5 (0.4) |
| 25% WF | 199.6 (16) | 1.23 (0.10) | 2 | 64 (2) | 30.8 | 1.9 (0.3) | 20.6 (2.6) | 2.0 (0.2) |
| 37% WF | 161.3 (6.5) | 1.21 (0.05) | 7 | 55 (2) | 30.0 | 2.9 (0.9) | 23.8 (2.8) | 2.2 (0.3) |
| 44% WF | 133.0 (6.5) | 1.21 (0.06) | 8 | 75 (4) | 22.4 | 3.4 (0.6) | 26.9 (2.4) | 2.7 (0.5) |
| 54% WF | 99.7 (4.5) | 1.22 (0.05) | 10 | 83 (1) | 25.3 | 3.6 (0.7) | 39.4 (6) | 2.5 (0.2) |
| 77% HP-WF | 180.4 (7.1) | 1.24 (0.05) | 13 | 34 (2) | 50.7 | 6.6 (1.8) | 82.1 (8.2) | 1.7 (0.4) |
| 27% MFLC | 103.6 (5.6) | 1.19 (0.06) | 6 | 77 (2) | 17.5 | 1.4 (0.1) | 43 (1.4) | 16.3 (2.3) |
| 35% MFLC | 97.6 (7.4) | 1.22 (0.09) | 6 | 69 (1) | 16.2 | 1.4 (0.3) | 53.2 (4.1) | 14.6 (0.5) |
| 41% MFLC | 61.9 (2.1) | 1.30 (0.04) | 1 | 46 (2) | 29.4 | 3.2 (0.4) | 61.1 (4.6) | 5.5 (0.8) |
| 50% MFLC | 56.0 (2.1) | 1.30 (0.05) | 3 | 55 (2) | 20.2 | 3.1 (0.3) | 62.0 (11.0) | 6.6 (1.3) |

Values in parentheses are standard deviations. Degree of crystallinity of c-PCL is calculated from first heating cycle of original samples (no thermal history removal), assuming melting enthalpy of 136.4 J/g for 100% crystalline PCL.

biocomposites with different reinforcement content. Strength and modulus data are provided as a function of fiber content in Fig. 4b. Compared with the modulus (0.5 GPa) and tensile strength (11.5 MPa) of neat c-PCL, properties are significantly improved. It is also apparent that MFLC provides much better strength reinforcement than WF at a comparable fiber content. In addition, MFLC biocomposites show substantially higher strain to failure compared to WF composites, which in combination with strain-hardening behavior leads to higher strength. The strain to failure is as high as 14–16% for compositions with 27 and 35% MFLC. The main reason is the small scale of MFLC nanofibers. For c-PCL/WF, the coarser structure (10 µm fiber diameter scale compared with 10–1000 nm nanofiber diameter scale for MFLC) leads to formation of defects and microcracks in the polymer matrix during deformation. This often takes place at voids or the fiber/polymer interface, and at much lower strain than for c-PCL/MFLC. The increase in c-PCL/WF biocomposites modulus with fiber content is consistent, but c-PCL/MFLC data shows significant scatter and effects are not consistent. One reason is the low crystallinity for some compositions, for instance c-PCL/MFLC with 50% MFLC. One may note that c-PCL/MFLC modulus data are lower than for comparable WF composites. For the c-PCL/MFLC with 27 and 35% MFLC, the PCL crystallinities (Table 1) are also lower than for comparable WF biocomposites, which may explain the difference.

Overall, high reinforcement content composites resulted in good mechanical properties. The c-PCL/MFLC41 biocomposite showed modulus above 3 GPa and ultimate strength above 60 MPa. This improvement extends the application range for PCL-based materials and may inspire investigations of other biocomposites from aliphatic polyesters (PBS, PLA). It is interesting to note that the wood fiber biocomposite, c-PCL/WF54 showed a higher modulus (3.6 GPa) than the comparable MFLC biocomposites and strength as high as 39.4 MPa. Since c-PCL is semi-crystalline and formed by in-situ polymerization, polymer properties are influenced by variations in gel content, average molar mass of extractable PCL fraction, and c-PCL crystallinity in different biocomposites.

The present PCL-based biocomposites show higher reinforcement content than in the large majority of previous studies and the wood fibers are largely undamaged which means mechanical properties are much improved, Fig. 4b. In most previous studies either chemical reinforcement modification, water-assisted melt compounding, or addition of a compatibilizer[57,67–70] is used to improve PCL-fiber stress transfer. Melt compounding results in mechanical damage to the fibers and fiber agglomerates are often present, particularly for nanocellulose or at high reinforcement content. The present use of a preformed WF or MFLC network makes high reinforcement content possible, results in well-preserved and well-dispersed fibers, and processing is direct and energy-efficient by oligomer impregnation followed by in-

situ polymerization of a degradable polymer matrix. The well-preserved structure of both types of reinforcements (WF and MFLC, Supplementary Fig. 30) makes a fair comparison possible of effects from reinforcement content and type, since the orientation distributions and fiber contents are similar.

To investigate the potential for even better mechanical properties, an already hot-pressed wood fiber sheet was used as reinforcement. This dense reinforcement was impregnated by c-PCL oligomers and polymerized. A biocomposite with 77 wt% HP-WF was obtained (c-PCL/HP-WF77), which showed the highest strength (82 MPa) and modulus (6.6 GPa) of all materials in Fig. 4b. This biocomposite also showed remarkable wet strength, modulus, and strain to failure of 50 MPa, 1 GPa, and 9.6%, respectively (Supplementary Fig. 35b). The main reason for these improvements is the high fiber content, although polymer matrix crystallinity is also increased, Fig. 3c. Note that the mechanical properties of c-PCL/HP-WF77 are much lower than those of neat HP-WF given the incorporation of a low-modulus polymer matrix; Supplementary Fig. 35b.

The DMTA data are presented in Supplementary Fig. 22c. The storage modulus improves with reinforcement content over the complete temperature range, and the relative increase is highest above $T_g$. The loss modulus shows a peak ~−45 °C, which is associated with the c-PCL glass transition. This is also the temperature at which both neat c-PCL and the biocomposites show softening with increased temperature. Note that c-PCL/MFLC27 has lower storage modulus than c-PCL/WF25 but also lower c-PCL matrix crystallinity (Table 1). The loss modulus peak moves to higher temperatures and broadens for the 41% MFLC biocomposite. With such high MFLC content of high specific surface area, a large fraction of the c-PCL matrix is in direct contact with MFLC surfaces. This may constrain molecular mobility of the c-PCL phase and lead to increased $T_g$, Supplementary Fig. 22c.

### Green chemistry and circular economy aspects

Unbleached kraft wood fibers are interesting not only because they are biobased, but also because they have high production yield and a history of minimal chemical treatment[54,55]. Key eco-indicators of high-lignin unbleached fibers such as cumulative energy demand (7–11 MJ/Kg), global warming potential (-0.2 Kg $CO_2$ eq/Kg), and water depletion (0.02–0.04 m³/Kg) demonstrate relatively low environmental impact of these fibers[55]. In this study, we developed a new green in-situ polymerization system. We strive for a green chemical process that makes efficient use of (ideally renewable) raw materials, reduces waste, and avoids the use of toxic and/or hazardous chemicals and solvents in the synthesis and application of chemical products[71]. Here, we calculated key green chemistry metrics i.e., atom economy, reaction mass efficiency, and environmental factor, to quantify the eco-friendliness of this chemical process.

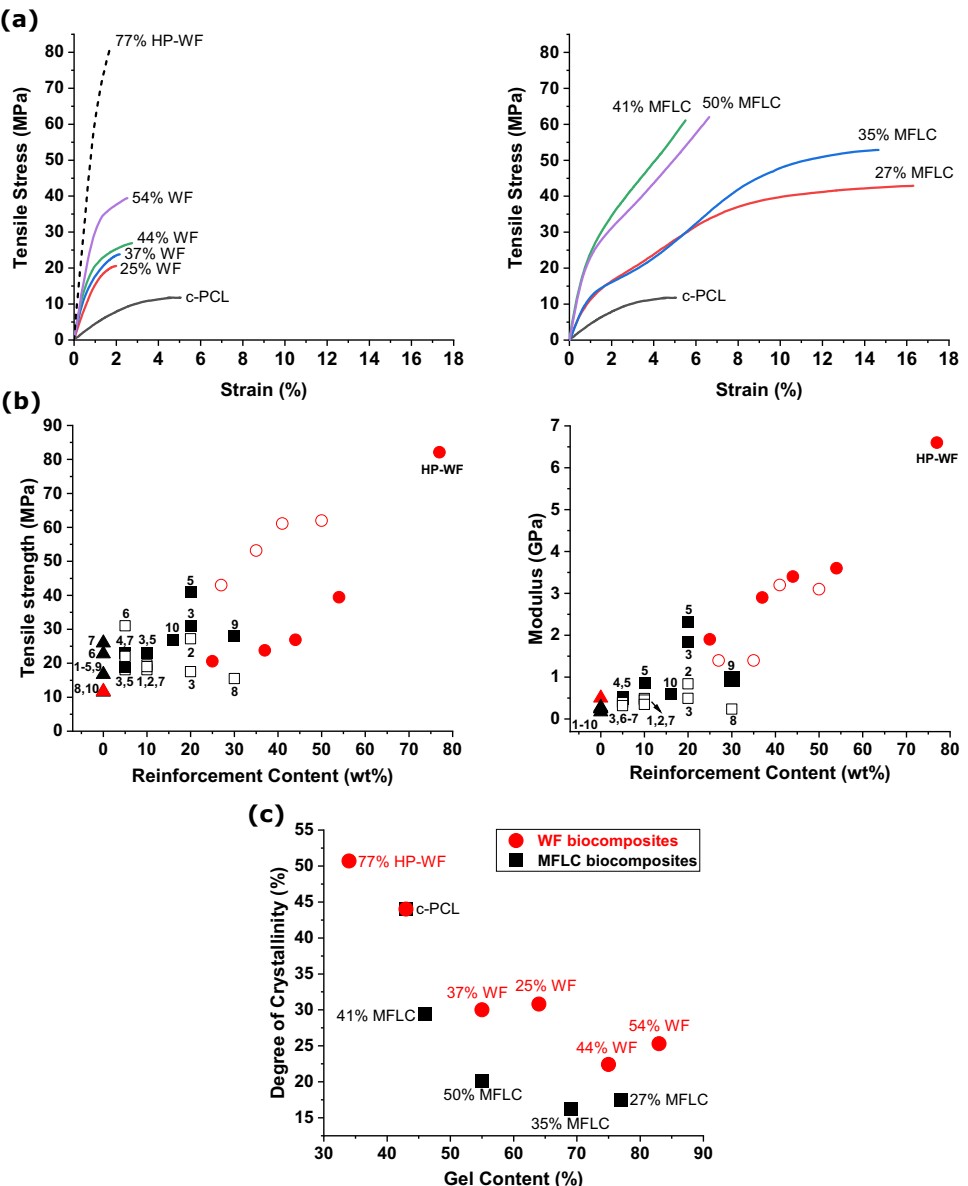

**Fig. 4 | Mechanical properties of the biocomposites at various reinforcement contents, together with information on the crystallinity and gel content of the polymer matrix. a** stress-strain curves of wood fibers (WF) and microfibrillated lignocellulose (MFLC) reinforced crosslinked polycaprolactone (c-PCL) composites. **b** Ultimate tensile strength and modulus versus lignocellulose reinforcement content in c-PCL composites. Square symbols: literature data from references 1:[67], 2:[68], 3:[77], 4:[57], 5:[69], 6:[70], 7:[78], 8:[79], 9:[80], and 10:[62]. Unfilled symbols: nanocellulose/microfibrillated cellulose composites; filled symbols: wood fiber composites; triangle: neat PCL; red color/circle symbols (no ref): data from the present study. **c** Degree of crystallinity vs gel content of WF and MFLC biocomposites (data from the first heating cycle without erasing processing history).

Trost's atom economy is a metric to show the reaction efficiency and is calculated as the number of atoms of reactants appearing in the product[71,72]. The atom economy of the whole series of reactions, including oligomer preparation, end-capping, and polyesterification/curing, is more than 99.5%, (note all c-PCL including gel and non-gel parts are desired, see calculation in Supplementary Discussion 3). The reaction mass efficiency is defined as the ratio of the actual mass of the desired product to the total mass of all reactants employed[73]. It considers both atom economy and chemical yield (e.g. excess reagents)[71,73]. Reaction mass efficiency of the whole series of reactions is 75%, and for only the in-situ polymerization/curing part is 100% (Supplementary Discussion 3).

Atom economy and reaction mass efficiency quantify the eco-friendliness of a reaction but not of a process. Neither of these metrics consider wastes generated by solvents and losses, whereas environmental factor does[71]. Sheldon's environmental factor (E factor) of a process is the ratio of the mass of waste per mass of product[71,74]. E factor for oligomer synthesis and end-capping is calculated ~5, and for polyesterification it is calculated to be 15 (Supplementary Discussion 3). Note, here, the aim is to create "model" biocomposites with homogeneous polymer distribution and low void content. Hence, solvent exchange and solvent-assisted impregnation are used to serve this purpose. In addition, in lab work, non-reacted reagents are discarded. These issues are the main culprits behind the E factor. However, for large-scale production, these issues are not so relevant, and E factor is potentially much lower. The solvents may not be necessary during infiltration as the molten oligomer at 140 °C displays viscosity values similar to commercial epoxy resin formulations (Supplementary Fig. 37), and non-reacted reagents can potentially be recycled. Therefore, from a

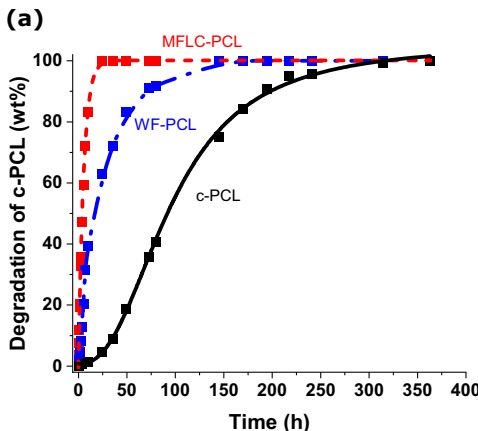

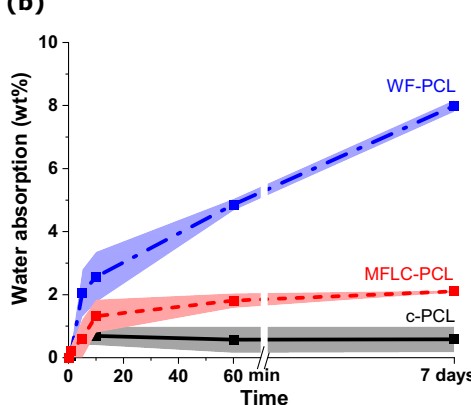

**Fig. 5 | Data on the progression of hydrolytic degradation and water absorption in biocomposites. a** Hydrolytic accelerated (alkaline) degradation of cross-linked polycaprolactone (c-PCL) homopolymer and biocomposites with 25% wood fibers (WF) and 27% microfibrillated lignocellulose (MFLC). **b** Water absorption of the homopolymer and biocomposites vs immersion time in water.

large-scale perspective, E factor for the polymerization part is almost zero (near ideal) since everything that remains in the system is desired, and there is no waste or byproducts.

Heating of biocomposites above 50 °C, the matrix melting point, resulted in a fluffy surface appearance dominated by fibers and the formation of substantial porosity in the material. This appearance is similar to that after in-situ polymerization. This porous material form may facilitate recycling and enhance composting/degradation rate. From an application perspective, however, it limits material usage to ambient conditions.

In a circular economy, degradability is critical for the end-of-life aspect of sustainable materials. PCL is particularly interesting since it can degrade in sea-water[75]. Accelerated tests under hydrolytic degradation conditions were performed to investigate reinforcement effects, since controlled degradation at industrial scale is an option to handle end-of-life challenges and promote a circular material economy. Degradation was monitored by ${}^1$H NMR under accelerated hydrolytic conditions using $D_2O$ at a pH of 12, leading to the formation of 6-hydroxycaproic acid sodium salt. The hydrolytic degradation rate was higher for the biocomposite compositions compared with neat c-PCL (Fig. 5a). In addition, the rate was very sensitive to the type of reinforcement, nanoscale MFLC or microscale WF. The c-PCL in the MFLC nanocomposite showed complete degradation after 25 h, compared to 140 h for the WF biocomposite. The data suggests that c-PCL in the fibrous lignocellulose networks degrades quickly under these conditions. The kinetics are faster for the c-PCL in more finely distributed MFLC nanofibers. The finely distributed MFLC network phase provides larger specific interfacial area. Since water prefers to be located in high energy interface regions, this also means that the specific surface area of c-PCL exposed to water becomes high. In contrast, degradation starts at the outer surface of neat c-PCL samples and moves inwards more slowly. Note biological degradation by microorganisms and their enzymes will progress by different mechanisms, which may change degradation kinetics compared with present results[76].

Water absorption data, Fig. 5b, shows low absorption for neat c-PCL, as expected. WF-based biocomposites absorb at a substantially higher rate than other biocomposites. This is interesting and related to lower swelling of the MFLC biocomposites due to well-bonded thin, nanoscale MFLC fibrils surrounded by c-PCL at a small scale. For wood fibers, there will be a contribution to high swelling from the thick (≈5 µm) and well-connected wood fiber cell walls, which provide a path for water diffusion or wicking. Overall, these results show that lignocellulosic fibers, besides acting as polymer reinforcement, can strongly contribute to matrix degradation behavior.

## Discussion

The plastics waste crisis highlights the need for a circular material economy. We also need materials that can compete with their petroleum-based counterparts in terms of both cost and properties. Many aliphatic polyesters are degradable but have insufficient mechanical properties to compete with petroleum-based bulk plastics. On the contrary, lignocellulosic fibers are strong but sensitive to moisture and difficult to shape with conventional processing techniques. Unfortunately, their combination has been limited to low fiber content composites due to processing limitations and synthetic problems related to moisture present in the fibers. We have developed a new reactive processing concept based on functionally balanced polycaprolactone (PCL) oligomers to address this. The PCL-oligomers enable direct in-situ polymerization within the fiber networks. The resulting PCL matrix is sparsely crosslinked, semicrystalline, and readily consolidated into low-porosity biocomposites of high reinforcement content. Biocomposites with more than 40 wt% reinforcement reach moduli as high as 3.1–6.6 GPa, and tensile strengths as high as 61–82 MPa, which exceeds typical data for commodity thermoplastics and is much higher than reported for previous wood fiber and nanocellulose PCL biocomposites. The high performance is based on the mechanically and chemically undamaged, well-dispersed reinforcement networks. The crosslinked PCL-matrix reduced moisture sensitivity and enabled biocomposite thermoforming for engineering applications. In addition, the materials are hydrolytically degradable, and the rate depends on reinforcement geometry (wood fibers or microfibrils).

High-lignin content wood fibers have a minimal environmental effect, and high-fiber content biocomposites produced by green reactive processing make the resulting biocomposites interesting for sustainable development. From a circular economy perspective, bio-based aliphatic polyesters are appealing as the polymer matrix for biocomposites. Polylactic acid is a particularly desirable candidate due to its excellent mechanical properties, competitive pricing, and wide availability. Overall, the present approach allows for improved mechanical performance and suggests eco-friendly degradable biocomposites from lignin-containing wood fibers as an alternative to plastics and polyethylene-based biocomposites in a circular materials economy.

## Characterization
### Gel content determination
A mixture of dimethylformamide/dichloromethane (70/30 wt%) was prepared and a piece of c-PCL homopolymer or biocomposite was added to the solvent mixture and let to be dissolved at room

temperature for 48 h under magnetic stirring. Note that we could not dissolve more PCL at more extreme conditions. Afterward, the solution was filtered off through a 0.65 μm PvDF membrane, and the dissolved part of the polymer, which passed through the membrane was measured as the filtrate dry weight. By knowing the initial composition of the materials i.e., c-PCL and the reinforcement content, the undissolved part of PCL was calculated as the gel content.

## Size exclusion chromatography

Size exclusion chromatography (SEC) was performed with a TOSOH EcoSEC HLC-8320GPC system equipped with an EcoSEC RI detector and three PSS PFG 5 μm columns (microguard, 100 Å, and 300 Å). A calibration curve was created using narrow, linear poly(methyl methacrylate) standards ranging from 700 to 2,000,000 g/mol. The typical sample concentration was 3 mg/mL and the flow rate was 1 mL/min at 30 °C. Corrections for flow-rate fluctuations were made using toluene as an internal standard. PSSWinGPC Unity software was used to process data. Each sample was analyzed twice and the average value was selected.

## Nuclear magnetic resonance

[1]H and [13]C NMR results were recorded at room temperature on a Bruker Avance III HD 400 MHz instrument with a BBFO probe equipped with a Z-gradient coil for structural analysis. Data were processed with MestreNova (Mestrelab Research) software using a 90° shifted square sine-bell apodization window, baseline and phase correction were applied in both directions.

## Fourier transform infrared

Fourier transform infrared (FTIR) spectra were obtained using an FTIR spectrometer (Perkin-Elmer, Spectrum 100) equipped with an attenuated total reflection (ATR) unit (Graseby Specac Ltd., England). Spectra were recorded with a resolution of 4 cm$^{-1}$ with eight scans in the range of 4000–600 cm$^{-1}$. For the comparison of the composites, the intensity of the hydroxyl stretching peak (3330 cm$^{-1}$) related to cellulose reinforcement was normalized to unity.

## Microscopy

A Hitachi S-4800 field-emission scanning electron microscope was used for imaging and morphological studies. Cross-sections of the films were prepared either by 10 min dipping in liquid nitrogen and cryo-fracturing or cutting using a UV KrF excimer laser (Lumonics, Canada). The laser had a wavelength of 248 nm, a frequency of 10 Hz, and energy of ~265 mJ . All samples were sputter-coated for 20–40 s with platinum-palladium using a 208HR Cressington Sputter Coater before the SEM study.

## Thickness and tensile test

The thickness of the samples was measured using an M201 structural thickness tester (TJT Teknik AB, Sweden). Strips of 6 mm × 45 mm were punched from the films and loaded in an Instron 5944 tensile machine (USA), equipped with a 500 N load cell and a video extensometer. The tests were conducted at 30 mm gauge length and a crosshead speed of 5 mm min$^{-1}$. The samples were conditioned for 48 h at 23 °C and 50% relative humidity. The wet tensile test was carried out on the samples immersed in water for 8 days.

## Dynamic mechanical analysis

The dynamic mechanical thermal properties of the composites were measured using a dynamic mechanical analyzer, DMA Q850 TA Instruments (New Jersey, USA). The measurements were performed using oscillation at a constant frequency of 1 Hz, amplitude of 10 μm, with a temperature ramp from −80 °C to 35 °C at a heating rate of 3 °C/min. The specimen dimensions were 6 mm × 45 mm, and the

gauge length was ~20 mm. A few measurements were carried out for each sample.

## Thermal characterization

Thermogravimetric analysis was performed via TGA Q5000 IR, TA Instruments. The samples were heated to 105 °C and pre-dried for 20 min, then were further heated to 800 °C (10 °C min$^{-1}$) in a nitrogen atmosphere (25 mL min$^{-1}$) to investigate the thermal stability of the samples.

Differential scanning calorimetry (DSC) (Q1000, TA Instruments, USA) was also carried out with a $N_2$ flow of 50 mL/min. Moisture was removed by pre-drying (105 °C, 20 min), the samples were heated to 120 °C and then quenched to −80 °C. Subsequently, the samples were heated from −80 °C to 150 °C (10 °C min$^{-1}$) to record possible glass transitions and melting temperatures. Data were evaluated using the Universal Analysis 2000 software (TA Instruments).

## Degradation study of the biocomposites

In the literature, there are examples of PCL accelerated degradation. However, our method to measure it is unique in its simplicity. The degradation of the pure c-PCL homopolymer and the WF and MFLC biocomposites were studied under static conditions by [1]H NMR. The chemical shift of the ε-protons of c-PCL is highly dependent on the chain-end or the repeating unit position. Therefore, a master solution of $D_2O$ at a pH of 12 was prepared with the addition of NaH to $D_2O$. After the complete reaction, benzyl alcohol was added as the internal standard. The degradation was started by adding 100 mg of materials into 1 ml of master solution in each nuclear magnetic resonance (NMR) tube and monitored through the increase in ε-protons, related to degraded PCL over time.

# Methods

## Materials and chemicals

Spruce wood chips provided by MoRe Research was used to prepare wood fibers and microfibrillated lignocellulose. ε-caprolactone (97%), glycerol (99.5%), succinic anhydride (99%), methanesulfonic acid (99%), dioxane (99.5%), ethanol (99.9%), acetone (99.5%), dimethylformamide (99.5%), dichloromethane (99.5%), sodium hydride (60%), and deuterium oxide (99.9%) from Sigma-Aldrich, Sweden were used. Tin (II) 2-ethylhexanoate (95%) from VWR was used.

## Preparation of wood fibers and fibrils

Unbleached pulp fibers with kappa 96 (16.9% lignin, 17.3% hemicellulose, 64.1% cellulose, and 1.7% extractives) were prepared from kraft pulping of spruce wood chips (18% effective alkali, 20% sulfidity, liquor to wood ratio of 4:1 and 90 min cooking at 160 °C). The wood fibers had an average length of 2.46 mm and average width of 39.1 μm (Supplementary Fig. 36 for full size distribution). Fiber fine (length < 0.1 mm) content, i.e. length-weighted average was 13.2%. The fibers were beaten and using a high-pressure microfludizer fibrillated into microfibrillated lignocellulose (MFLC) consisting of 44 wt% fibrils with nanosized (<100 nm) thickness and rest coarse fibrils with up to 1 μm thickness. All populations of MFLC fibrils (unfractionated) were used in this study.

## Functional ε−caprolactone-oligomer synthesis

The functional εCL oligomers were synthesized through ring-opening polymerization initiated by glycerol and the outcome was partially end-capped by succinic anhydride. In an example reaction, glycerol (0.48 g, 5.2 mmol, 1 equiv.) was dissolved in 30 ml of dry dioxane in a 100 ml dried round-bottom flask equipped with a magnetic stirrer. After dissolution, ε-caprolactone (8.3 g, 73 mmol, 14 equiv.) and the catalyst methanesulfonic acid (0.07 g, 0.73 mmol, 0.14 equiv.) were added. The polymerization was left to proceed for 24 h at ambient temperature. After 24 h, succinic anhydride (1.55 g, 15.5 mmol, 3 equiv.)

was added to the reaction mixture. After 72 h, the end-group conversion was roughly 50%, according to $^1$H NMR. At this point, the reaction was terminated by 500 ml of water and the reaction mixture was decanted. The mixture was placed in the fridge overnight and filtered the next day. The product was air-dried for 5 days, and the isolated yield was calculated 75%.

### In-situ polymerization of oligomers, and preparation of biocomposites

Dilute aqueous suspension of fibers or MFLC was vacuum-filtrated. Afterward, the formed wet networks (ca 85% water) were subjected to solvent exchange to ethanol and then acetone (a few times), respectively. The solvent exchange is performed in order to reduce moisture content (though a small amount may remain). The caprolactone three-arm oligomer was dissolved in acetone and added to fiber/fibril networks. The mixtures were kept at room temperature to impregnate for a few hours. Then, stannous octoate as the catalyst (0.5 mol%) was added to the mixtures, and the reactions were performed at 140 °C for 14 h. A reference of pure c-PCL was also prepared with the same procedure. Note that the aim is to create "model" biocomposites with homogeneous polymer distribution and low void content. Hence, solvent exchange and solvent-assisted impregnation are used to serve this purpose. After polymerization, the composites were hot-pressed at 40–50 MPa and 120 °C for 5 min. To create a homopolymer sheet, just 1 MPa pressure and 60 °C for 3 min was enough. For the re-shaping experiment, an already hot-pressed film was re-hot-pressed into a 3D structure.

### Data availability

The datasets generated and/or analyzed during the current study are supplied in the supplementary information. If additional data or information is sought, this will be provided by the corresponding author upon request.

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

## Acknowledgements

E.O. would like to thank Céline Montanari for help with SEM, Henrik Petterson for help with the UV excimer laser and Dr. Pratick Samanta for discussion on DSC data. E.O. would like to thank Dr. Joseph G. Manion for the beaker artwork design (blender file). This work is supported by Swedish Foundation for Strategic Research (grant FID15-0115 TL and LAB), KAW Biocomposites project (grant 2018.0451 LAB), and funding from Formas – a Swedish Research Council for Sustainable Development (Re-Design Plastic, 2020-01696 PO).

## Author contributions

Conceptualization by E.O., P.O., and L.A.B.; experimental design by E.O., P.O., and L.A.B.; experimental work and data analysis by E.O. and P.O.; data curation and visualization by E.O.; writing—draft preparation by E.O. and P.O.; review and editing by L.A.B. and T.L.; project administration by T.L. and L.A.B.; funding acquisition by T.L. and L.A.B. All authors have read and agreed to the published version of the manuscript.

## Funding

## Competing interests

The authors declare no competing interests.
