## [Peer Review File · Nature Communications]

Highly reinforced and degradable lignocellulose biocomposites by polymerization of new polyester oligomersReviewers' Comments:

Reviewer #1:

Remarks to the Author:

The authors report lignocellulose biocomposites formulated via hot-pressing. Overall, the manuscript has some nice features that seem scalable and provide excellent mechanical properties. In the current form, this manuscript would be better suited for a more specialized journal and needs some concerns addressed before it is ready for publication.

1) There is a disconnect between the title, which contains "Green Polymerization" and the discussion of sustainability in the introduction. While having "Green Polymerization" in the title is reasonable, the authors do not give any metrics or life cycle analysis to quantify the degree of sustainability. Certainly the authors have elements that could be viewed as sustainable, such as bio-based monomers, use of catalysis, and low-temperature curing, but upon closer inspection, this manuscript is mainly bio-based and not sustainable. In other words, while the authors use catalysis, the catalyst loading (0.5 -1.0 mol%) is high and Sn(Oct)₂ has questionable toxicity. While some of the materials, such as lignocellulose are bio-based, caprolactone, dioxane, DMF, and dichloromethane from Sigma-Aldrich are probably petroleum-based. Furthermore, the amount of chemical waste produced in Figure 1a is quite high and reduces the degree of sustainability. The authors should calculate Roger Sheldon's E factor, which is the ratio of waste to product, to better understand their processes. A quick estimate of Figure 1a indicates the E factor is > 40.

2) A recent article (ACS Macro Lett. 2021, 10, 1, 41–53) states that "comprehensive literature survey reveals that there has been an increased focus on "sustainable polymers" in recent years, but most papers focus on biomass-derived feedstocks." The authors of this manuscript have made a similar assumption and mistake "bio-based" for "sustainable."

3) The authors state that "biodegradation is often the best way of final disposal for a "sustainable" biocomposite" and cite reference 10, which is 27 years old, and reference 11 which deals with flexible electronics. Although biodegradation is appealing, this approach is too slow to deal with the global plastic accumulation. I recommend the authors put their work in the context of other perspectives (Nature Climate Change volume 9, pages 374–378 (2019)) which advocate for recycling or circular economy.

4) The accelerated degradation studies in this manuscript are interesting but not very useful. While accelerated studies do provide insight into whether degradation is possible, the timescale for degradation varies widely depending on conditions (Angew.Chem.Int.Ed.2019,58,50–62) and soil is expected to be different than seawater. I appreciate the authors cite references 66 and 67 and also state "biological degradation by microorganisms and their enzymes will progress by different mechanisms, which may change degradation kinetics compared with present results." However, since biodegradation represents an important aspect of this manuscript, non-accelerated degradation studies in soil and seawater are crucial. In addition, since the authors use Sn(Oct)₂ to polymerize caprolactone, it will be crucial to determine if Sn(Oct)₂ inhibits biodegradation.

4) Since the polymerization of caprolactone with MSA is common (Macromolecules 2013, 46, 4354–4360), please clarify what aspect of the polyester synthesis is new. I also see literature examples that polymerize caprolactone with Sn(Oct)₂ and subsequently functionalize PCL with succinic anhydride (European Polymer Journal 45 (2009) 557–564).

Reviewer #2:

Remarks to the Author:

The manuscript is innovative and offers a large body of results from different techniques to contribute

to the analysis of the behavior of these new composites. The comments listed below are mainly directed to the clarification of some of the trends observed (or the lack of one clear trend) in the results.

Line 73: "MFLC" The complete name is not included here (first time that it appears in the text), but in line 87 ("Microfibrillated lignocellulose (MFLC)"). Revise.

Add a comment regarding the chemical composition of WF and MFLC, are they similar? or they differ in the content of lignin, hemicellulose, waxes...? Difference in surface chemistry may contribute (additionally to the difference in specific surface area) to the different degree of interaction of the two types of lignocellulosics considered with the PCL-based polymer.

There is a clear trend of crystallinity decreasing with increasing gel content, which is understandable because of the increasing reduced mobility of the polymer as the gel content increases. However, there is not a trend (at least not a clear monotonous trend) of the gel content with the increasing amount of filler (or the crystallinity with the filler concentration). Could you advance an explanation for this?

In Page 15, the label in the "y"-axis of Figure 4a has a typographical error: "Degaraded". Besides, it should not be labeled as the percentage of degraded cPCL, since at time = 0 it has not been degraded and the plot would begin at 0%. Instead, it shows the remaining undegraded percentage of c-PCL.

MFLC-PCL degrades much faster than the WF-PCL or the c-PCL, however it absorbs much less water than WF-PCL. Can you offer an explanation of the reason why the hydrolytic degradation occurs more slowly in the composites capable of absorbing more water (WF-PCL)?

While the MFLC are non porous micro/nanofibers that interact with the polymer via their external surface, the WF are whole fibers that, besides their external surface, have internal cell volumes (lumen, for instance) that can be filled with the oligomers before reaction. Is it possible that polymer trapped inside the WF is yet another reason for the difference observed between the composites prepared from the two reinforcements?

Also, while short polymer chains diffuse more rapidly and can reach (or leave) the surface of the reinforcements with higher velocity, the long polymer chains are slower, but once that they develop a strong interaction with a surface, they are also more difficult to detach. This may also be a factor to consider in the differences of extracted polymer (dissolved polymer).

Reviewer #3:

Remarks to the Author:

Question 1: Many kinds of polyesters in life are engineering plastics with excellent performance and wide use, such as polyethylene glycol terephthalate (PET), polybutylene terephthalate (PBT). Which polyesters have low mechanical properties on page 2, lines 47-49?

Question 2: The sentence pointed out the shortcomings of the melt-processing method on page 2, lines 41-43. While, some other shortcomings were pointed out on page 4, lines 90-92. Why do they have to be explained separately?

Question 3: What are the in situ polymerization methods of cellulose, except ROP reaction? ATRP may be one of them.

Question 4: "PCL is investigated as a model polymer as candidate replacement of polyethylene (PE) and polypropylene (PP) biocomposite matrices." What other polyesters do you think are similar to PCL

and can be served as a model polymer? In addition, it is not logical for PCL to replace PE and PP biocomposite matrices.

Question 5: In figure 1(a), the molecular formula of the product showed that the PCL oligomers were terminated by carboxyl groups. But the picture on the right showed that the oligomers have both hydroxyl and carboxyl ends.

Question 6: On page 7, lines 154-155 (page 8, lines 177-178), have the lengths and diameters of WF and MFLC used in this paper been tested? The datas given in this paper were those in the literature.

Question 7: "Fig. 1a and b" on page 8, lines 177 may be wrong.

Question 8: Is the porosity (density) of the material closely related to the hot pressing pressure? Larger pressure (40-50 MPa) was used in this paper.

Question 9: Is there a comparison of mechanical properties between the hot-pressed wood fiber sheet and c-PCL/HP-WF77?

Question 10: Did the degradation test reflect true degradation? What is the ordinate of Figure 4(a)?

We have edited the manuscript, and you will find a point-by-point response to the reviewers' comments on the following pages.

Reviewer #1

The authors report lignocellulose biocomposites formulated via hot-pressing. Overall, the manuscript has some nice features that seem scalable and provide excellent mechanical properties. In the current form, this manuscript would be better suited for a more specialized journal and needs some concerns addressed before it is ready for publication.

1) There is a disconnect between the title, which contains "Green Polymerization" and the discussion of sustainability in the introduction. While having "Green Polymerization" in the title is reasonable, the authors do not give any metrics or life cycle analysis to quantify the degree of sustainability. Certainly the authors have elements that could be viewed as sustainable, such as bio-based monomers, use of catalysis, and low-temperature curing, but upon closer inspection, this manuscript is mainly bio-based and not sustainable. In other words, while the authors use catalysis, the catalyst loading (0.5 -1.0 mol%) is high and Sn(Oct)₂ has questionable toxicity. While some of the materials, such as lignocellulose are bio-based, caprolactone, dioxane, DMF, and dichloromethane from Sigma-Aldrich are probably petroleum-based. Furthermore, the amount of chemical waste produced in Figure 1a is quite high and reduces the degree of sustainability. The authors should calculate Roger Sheldon's E factor, which is the ratio of waste to product, to better understand their processes. A quick estimate of Figure 1a indicates the E factor is > 40.

Response:

We have now edited the title and introduction. Please see the below clarification.

- We considered our method to be a model polymerization system to demonstrate its feasibility, and certain aspects require refinement. For example, Sn(Oct)₂ catalysis should be replaced with a more efficient and environmentally friendly catalysis system.
- In a closed system, dioxane is employed for oligomer synthesis. As a result, solvent recycling could be considered if this synthesis method is scaled up.
- Although ε-caprolactone monomer is from petroleum resources, the resulting c-PCL is hydrolytically degradable, making it particularly desirable for chemical recovery and the circular economy.
- DMF and dichloromethane were used to quantify gel content and not in the polymerization process.

Comments related to sustainability and biobased are addressed in the next question. The manuscript is modified extensively and accordingly. We have added several parts to explain green chemistry in the Main, discussion, and SI.

SI:

"Green chemistry metrics

Atom economy:

Trost's atom economy is a metric to show the reaction efficiency and related to the number of atoms of reactants appearing in the product^{1,2}, calculated as

Eq. S4.

$$\text{Atom economy} = \frac{\text{molecular mass of desired product}}{\text{molecular masses of reactants}}; \text{ (In the stoichiometric equation)}$$

We have 3 different reactions: ROP of ϵ -caprolactone, end-capping, and polyesterification reactions.

ROP reaction:

End-capping reaction:

Atom economy of ROP and end-capping for the functionally balanced oligomer (actually a mixture of OligoA and OligoB)

$$\text{OligoA} = 1788 / (92 + 14 * 114 + 100) = 1788 = 100\%$$

$$\text{OligoB} = 1788 / (92 + 14 * 114 + 2 * 100) = 1888 = 100\%$$

Polyesterification reaction:

$$\text{Atom economy of polyesterification} = (\text{molecular mass of ROOR}') / (\text{sum of molecular masses of ROOH} + \text{R}'\text{OH}) > (1788 * 2 - 18) / (1788 * 2) > 99.5\%$$

In polyesterification, water molecule is released. Esterification of smallest oligomers make a di-oligomer with the least possible atom economy of polyesterification, since the product has the least possible mass (relative to mass of undesired product, i.e. water).

Note that desired product is c-PCL, both the gel and non-gel fractions. Therefore, atom economy of the whole series of reactions from oligomer synthesis to polyesterification is > 99.5%.

Reaction mass efficiency:

Reaction mass efficiency is defined as the ratio of the actual mass of the desired product to the total mass of all reactants employed³. It considers both atom economy and chemical yield (e.g. excess reagents)^{1,3}.

Eq. S5.

$$\text{Reaction mass efficiency} = \frac{\text{mass of desired product}}{\text{masses of reactants}}; \text{ (In practice)}$$

$$\text{Reaction mass efficiency of ROP and end-capping} = 1500 / (92 + 114 * 14 + 100 * 3) = 75\%$$

$$\text{Reaction mass efficiency of polyesterification} = 100\%$$

The reaction mass efficiency of the whole series of reactions from oligomer preparation to polyesterification is 75% (75% * 100%), and for only the in-situ polymerization/curing part is 100%.

Sheldon's environmental factor (E factor):

Sheldon's E-factor of a process is defined as Eq. S6.

$$E - \text{factor} = \frac{\text{mass of waste}}{\text{mass of product}}$$

E factor for the synthesis of the selected oligomers (Gly-PCL₁₄-SA_{1.5} with 0.5 mol% catalyst) is calculated:

Waste [30 ml of dioxane (30.9 g)+ 0.5*methanesulfonic acid (0.5*0.07 g)+ 0.5*succinic anhydride (0.5*1.55 g)] / product [caprolactone (8.3 g)*yield(~0.75)] => Sheldon's E factor (30.9+0.03+0.77 g)/(6.22 g) = 5.1

Note that in the scaled-up synthesis of oligomers, the solvent is not discarded but recycled. Also, the yield would be much higher than 75% since oligomer isolation procedure will be optimized. Therefore, Sheldon's E factor would be much lower than 5.

All the waste in the polymerization system (from oligomers) is related to ethanol and acetone losses in solvent exchange and impregnation. Each time solvent equal to 2.5 times the weight of the product is used for solvent exchange, and acetone equal to 5 times the weight of the product is used for impregnation. One time, ethanol and 3 times acetone were exchanged, and then fresh acetone was used for impregnation. Hence, Sheldon's E factor for the in-situ polymerization part is approximately 15,

$$E\text{-factor} = 1*2.5+3*2.5+1*5 = 15$$

Note here the aim is to create "model" biocomposites with homogeneous polymer distribution and low void content. Hence, solvent exchange and solvent-assisted impregnation are used to serve this purpose. However, for large-scale production, the use of solvents is not needed (at 140 °C the oligomers are liquid and can diffuse to fiber networks). Therefore, from a large-scale perspective, Sheldon's E factor for the polymerization part is almost zero (near ideal) since everything that remains in the system is desired, and there is no loss or byproducts. However, quality may, to some extent, be compromised.

".

2) A recent article (ACS Macro Lett. 2021, 10, 1, 41–53) states that "comprehensive literature survey reveals that there has been an increased focus on "sustainable polymers" in recent years, but most papers focus on biomass-derived feedstocks." The authors of this manuscript have made a similar assumption and mistake "bio-based" for "sustainable."

Response:

The phrases "biobased" and "sustainable" are now used more cautiously in the manuscript.

We previously showed (referenced in the manuscript) that important eco-indicators (cumulative energy demand, global warming potential, and water depletion) show low values for the high-lignin kraft wood fibers used in this study). Here we investigated and developed an in-situ polymerization system and showed the green chemistry aspects of this chemical process and quantified environmental impact using key green chemistry metrics (atom economy, reaction mass efficiency, and E factor).

Page 4:

“The investigated biobased unbleached kraft fibers show higher yield, lower energy demand, carbon footprint, and are subjected to less severe chemical treatment than cellulosic wood pulp fibers of low lignin content^{4,5}. Important eco-indicators of high-lignin unbleached fibers such as cumulative energy demand (7-11 MJ/Kg), global warming potential (~0.2 Kg CO₂ eq/Kg), and water depletion (0.02-0.04 m³/Kg) values show low environmental impact and are typically 30% lower than bleached fibers⁵. Furthermore, they show little mechanical damage and favorable compatibility with non-polar polymers. MFLC investigated here is also characterized by high yield, reduced water sensitivity, and better compatibility with “hydrophobic” polymers compared to higher-purity microfibrillated cellulose⁴⁻⁶, although they are not showing interesting cumulative energy demand (>100 MJ/Kg)⁴. Note that MFLC is used for mechanical properties comparison with fibers, not environmental aspects. In this study, we developed an in-situ polymerization system and showed the greenness of this chemical process and its low environmental impact using key green chemistry metrics (atom economy, reaction mass efficiency, and E factor).”

3) The authors state that "biodegradation is often the best way of final disposal for a “sustainable” biocomposite" and cite reference 10, which is 27 years old, and reference 11 which deals with flexible electronics. Although biodegradation is appealing, this approach is too slow to deal with the global plastic accumulation. I recommend the authors put their work in the context of other perspectives (Nature Climate Change volume 9, pages 374–378 (2019)) which advocate for recycling or circular economy.

Response:

This is very helpful, and we have modified the manuscript accordingly with a stronger focus on circular material systems as an ideal option to address the challenges of plastics waste. The text in the manuscript has been modified accordingly and extensively.

Main and Discussion section is rewritten with circular economy perspective. The previous End-of-life section is completely rewritten, and the section is renamed “Green chemistry and circular economy aspects”. New references including the following references has been added to the manuscript:

- Zheng, J. & Suh, S. Strategies to reduce the global carbon footprint of plastics. *Nature Climate Change* **9**, 374-378, doi:10.1038/s41558-019-0459-z (2019).
- Cabernard, L., Pfister, S., Oberschelp, C. & Hellweg, S. Growing environmental footprint of plastics driven by coal combustion. *Nature Sustainability* **5**, 139-148, doi:10.1038/s41893-021-00807-2 (2022).
- Mohanty, A. K. *et al.* Sustainable polymers. *Nature Reviews Methods Primers* **2**, 46, doi:10.1038/s43586-022-00124-8 (2022).
- Hong, M. & Chen, E. Y. X. Chemically recyclable polymers: a circular economy approach to sustainability. *Green Chemistry* **19**, 3692-3706, doi:10.1039/C7GC01496A (2017).
- Mohanty, A. K., Vivekanandhan, S., Pin, J.-M. & Misra, M. Composites from renewable and sustainable resources: Challenges and innovations. *Science* **362**, 536-542, doi:doi:10.1126/science.aat9072 (2018).
- Stahel, W. R. The circular economy. *Nature* **531**, 435-438, doi:10.1038/531435a (2016).
- Closing the plastics loop. *Nature Sustainability* **1**, 205-205, doi:10.1038/s41893-018-0075-3 (2018).
- Rahimi, A. & García, J. M. Chemical recycling of waste plastics for new materials production. *Nature Reviews Chemistry* **1**, article 0046, doi:10.1038/s41570-017-0046 (2017).

4) The accelerated degradation studies in this manuscript are interesting but not very useful. While accelerated studies do provide insight into whether degradation is possible, the timescale for degradation varies widely depending on conditions (Angew.Chem.Int.Ed.2019,58,50–62) and soil is expected to be different than seawater. I appreciate the authors cite references 66 and 67 and also state "biological degradation by microorganisms and their enzymes will progress by different mechanisms, which may change degradation kinetics compared with present results." However, since biodegradation represents an important aspect of this manuscript, non-accelerated degradation studies in soil and seawater are crucial. In addition, since the authors use Sn(Oct)₂ to polymerize caprolactone, it will be crucial to determine if Sn(Oct)₂ inhibits biodegradation.

Response:

We agree that many factors can influence biodegradation; environment, humidity, temperature, and residual catalyst, to name a few. In the new version of the manuscript, we have embraced the reviewer's suggestion using the perspective of a circular material economy. Still, Sn(Oct)₂ may pose a challenge due to toxicity, and future research will try to target this by creating more selective and benign linking chemistry.

To clarify, multiple passages have been added to the new version of the manuscript as mentioned in response to the previous question.

4) Since the polymerization of caprolactone with MSA is common (Macromolecules 2013, 46, 4354–4360), please clarify what aspect of the polyester synthesis is new. I also see literature examples that polymerize caprolactone with Sn(Oct)₂ and subsequently functionalize PCL with succinic anhydride (European Polymer Journal 45 (2009) 557–564).

Response:

ε-caprolactone (εCL) polymerization with methane sulfonic acid (MSA) is a known reaction. This reaction was only used for synthesis of the multifunctional oligomers. The novelty resides in the second step utilizing these oligomers for in-situ polymerization within the fiber network. The reason for using MSA as a catalyst instead of Sn(Oct)₂ for the oligomer synthesis is explained below.

- 1) MSA is highly active for the polymerization of εCL, enabling the polymerization at ambient temperature
- 2) The cationic polymerization method of εCL with MSA enables us to ensure that after end-capping with succinic anhydride (SA), we have the carboxylic acid in the R-COOH form to promote the condensation step.
- 3) Acid-catalyzed ring-opening of SA is comparably slower than for εCL. As such, we reduce the degree of transesterification while simultaneously enabling high control over the end-group conversion to achieve functionally balanced oligomers.

The new version of the manuscript has clarified this point more in detail, page 5.

"MSA is employed as a catalyst because it is highly active for the polymerization of CL, allowing for polymerization at ambient temperature. Acid-catalyzed ring-opening of SA is comparably slower than for εCL, so that the degree of transesterification is diminished while simultaneously enabling high control over the end-group conversion (to R-COOH) to produce functionally balanced oligomers"

Reviewer #2

The manuscript is innovative and offer a large body of results from different techniques to contribute to the analysis of the behavior of these new composites. The comments listed below are mainly directed to the clarification of some of the trends observed (or the lack of one clear trend) in the results.

Line 73: "MFLC" The complete name is not included here (first time that it appears in the text), but in line 87 ("Microfibrillated lignocellulose (MFLC)"). Revise.

Response:

Thank you for the comment; it is now fixed in the manuscript (page 4).

Add a comment regarding the chemical composition of WF and MFLC, are they similar? or they differ in the content of lignin, hemicellulose, waxes...? Difference in surface chemistry may contribute (additionally to the difference in specific surface area) to the different degree of interaction of the two types of lignocellulosics considered with the PCL-based polymer.

Response:

Thank you for the useful comment. They have similar composition since we made MFLC from the same wood fibers and we keep all the fibrillation products as MFLC. In methods: preparation of wood fibers and fibrils (page 19), we have explained that the composition is the same and we showed the contents. Also, in Lines 179-180 and 201-202, we have now added "networks (wood fibers or MFLC with the same compositions and sources)" and "Therefore, the main differences are related to size, specific surface area and intrinsic mechanical properties."

In the manuscript we have also added descriptions about influence of trapped or bonded polymer inside the lumen (pages 10 and 12).

There is a clear trend of crystallinity decreasing with increasing gel content, which is understandable because of the increasing reduced mobility of the polymer as the gel content increases. However, there is not a trend (at least not a clear monotonous trend) of the gel content with the increasing amount of filler (or the crystallinity with the filler concentration). Could you advance an explanation for this?

Response:

Yes, there is no monotonous trend of rising gel content (or reducing crystallinity) with increased reinforcement. We do not fully grasp the reasons since the system is complicated, and the polymer gel in this work can be grafted c-PCL molecules as well as large, cross-linked c-PCL molecules that are influenced by reinforcement characteristics. However, c-PCL in biocomposites definitely has a higher gel content and lower crystallinity than pure c-PCL. In page 12, we have further clarified this matter.

"c-PCL in biocomposites has a higher gel content and lower crystallinity than pure c-PCL. One reason is that oligomers react with the carboxyl and hydroxyl groups of the reinforcement surfaces, increasing the fraction of gel content. There is, however, not a clear monotonous trend of rising gel content (or reduced crystallinity) with lignocellulose content due to the complexity of the system. In wood fiber

biocomposites, increasing fiber content generally results in higher gel content. One reason may be c-PCL trapped or bonded inside wood fibers or in lumen region."

In Page 15, the label in the "y"-axis of Figure 4a has a typographical error: "Degaraded". Besides, it should not be labeled as the percentage of degraded cPCL, since at time = 0 it has not been degraded and the plot would begin at 0%. Instead, it shows the remaining undegraded percentage of c-PCL.

Response:

Absolutely. We have corrected the plot (Page17).

MFLC-PCL degrades much faster than the WF-PCL or the c-PCL, however it absorbs much less water than WF-PCL. Can you offer an explanation of the reason why the hydrolytic degradation occurs more slowly in the composites capable of absorbing more water (WF-PCL)?

Response:

We have now explained these phenomena in a better way (page 17). c-PCL hydrolysis begins at water-exposed surfaces, and MFLC biocomposites have more interfacial specific area. Furthermore, we explained the lower water absorption of MFLC-PCL with a tight and well-bonded MFLC network with limited swelling potential. WF-PCL, on the other hand, is likely to be influenced by thick (5 μ m) fiber cell walls, which provide a channel for water diffusion or wicking.

Page 17: "The finely distributed MFLC network phase provides larger specific interfacial area. Since water prefers to be located in high energy interface regions, this also means that the specific surface area of c-PCL exposed to water becomes high. In contrast, degradation starts at the outer surface of neat c-PCL samples and moves inwards more slowly.

Water absorption data, Fig. 4b, shows low absorption for neat c-PCL, as expected. WF-based biocomposites absorb at a substantially higher rate than other biocomposites. This is interesting and related to lower swelling of the MFLC biocomposites due to well-bonded thin, nanoscale MFLC fibrils surrounded by c-PCL at a small scale. For wood fibers, there will be a contribution to high swelling from the thick (\approx 5 μ m) and well-connected wood fiber cell walls, which provide a path for water diffusion or wicking."

While the MFLC are non porous micro/nanofibers that interact with the polymer via their external surface, the WF are whole fibers that, besides their external surface, have internal cell volumes (lumen, for instance) that can be filled with the oligomers before reaction. Is it possible that polymer trapped inside the WF is yet another reason for the difference observed between the composites prepared from the two reinforcements?

Response:

We added the explanation below to the manuscript on page 10. "Fig. 2d also shows c-PCL is also distributed in the wood fiber lumen region (inside the cell wall), also confirmed by Figure S26. Micro- and nanostructural differences between WF and MFLC biocomposites influence the reinforcement mechanisms."

And the point below on page 10:

"Polymer trapped or bonded inside the wood fiber cell wall or lumen may contribute to the higher gel content of c-PCL/WF44 compared to c-PCL/MFLC41."

Also, while short polymer chains diffuse more rapidly and can reach (or leave) the surface of the reinforcements with higher velocity, the long polymer chains are slower, but once that they develop a strong interaction with a surface, they are also more difficult to detach. This may also be a factor to consider in the differences of extracted polymer (dissolved polymer).

Response:

We added this explanation to the manuscript (page 10).

"The gel content was defined as the fraction of c-PCL that could not be dissolved (see Characterization section). Shorter polymer chains are readily soluble, whereas higher molar mass molecules have lower solubility. Furthermore, polymer chains strongly linked to the reinforcement cannot be extracted."

Reviewer #3

Question 1: Many kinds of polyesters in life are engineering plastics with excellent performance and wide use, such as polyethylene glycol terephthalate (PET), polybutylene terephthalate (PBT). Which polyesters have low mechanical properties on page 2, lines 47-49?

Response:

Thank you for the comment. We were referring to aliphatic polyesters. For clarification, the text (now on page 3) has been changed to "Aliphatic polyesters are interesting in this context as they are degradable to non-hazardous products in industrially relevant environments^{7,8}. A challenge is that most semi-crystalline aliphatic polyesters have low glass transition temperature T_g , which translates into low modulus and tensile strength⁹."

Question 2: The sentence pointed out the shortcomings of the melt-processing method on page 2, lines 41-43. While, some other shortcomings were pointed out on page 4, lines 90-92. Why do they have to be explained separately?

Response:

We have now combined these points (page 2).

Question 3: What are the in situ polymerization methods of cellulose, except ROP reaction? ATRP may be one of them.

Response:

Many different types of polymerization methods could be used for in-situ polymerization in fiber networks, and ATRP is one of them. For clarification, the following sentences and references have been added to the new version of the manuscript, page 3.

“In order to reach high fiber content, in-situ polymerization within a pre-formed fiber network is a feasible approach. In-situ polymerization of non-degradable polymers in cellulose fiber networks has been carried out through free radical polymerization^{10,11}, controlled radical polymerization¹², and ring-opening metathesis polymerization¹³. There is, however, little work using degradable polymers for high fiber content composites. The reason is that degradable polymer matrix synthesis is often based on ring-opening polymerization (ROP) of cyclic monomers¹⁴⁻¹⁸.”

Question 4: “PCL is investigated as a model polymer as candidate replacement of polyethylene (PE) and polypropylene (PP) biocomposite matrices.” What other polyesters do you think are similar to PCL and can be served as a model polymer? In addition, it is not logical for PCL to replace PE and PP biocomposite matrices.

Response:

Various biobased aliphatic polyesters that are attractive from a circular economy standpoint include poly (lactic acid), poly (butylene succinate), poly (ethylene adipate), poly (lactic-co-glycolic acid), and other copolymers. Certainly, PCL-fiber biocomposites are not perfect replacements for polyolefin products in terms of cost, chemical durability, shaping capability, and hydrophobicity. So, on the manuscript pages 4 and 18, we corrected and clarified our purpose as follows.

Page 4: “PCL is investigated as a model polymer representing soft, low T_g semicrystalline and compostable aliphatic polyesters as candidate replacement of polyethylene (PE) and polypropylene (PP) biocomposite matrices. Although neat, unmodified PCL shows low mechanical properties, the present wood fiber biocomposites with crosslinked, caprolactone-based c-PCL matrix show better mechanical properties than previous reports for biocomposites with aliphatic polyester matrix,¹⁹⁻²³ since the processing concept allows high content of well-dispersed, efficient network reinforcements. Chemical durability, shaping capability, and hydrophobicity of c-PCL biocomposites, on the other hand, are not competing with polyolefins.”

Page 18: “High-lignin content wood fibers have a minimal environmental effect, and high-fiber content biocomposites produced by green reactive processing make the resulting biocomposites interesting for sustainable development. From a circular economy perspective, biobased aliphatic polyesters are appealing as the polymer matrix for biocomposites. Polylactic acid is a particularly desirable candidate due to its excellent mechanical properties, competitive pricing, and wide availability. Overall, the present approach allows for improved mechanical performance and suggests eco-friendly degradable biocomposites from lignin-containing wood fibers as an alternative to plastics and polyethylene-based biocomposites in a circular materials economy.”

Question 5: In figure 1(a), the molecular formula of the product showed that the PCL oligomers were terminated by carboxyl groups. But the picture on the right showed that the oligomers have both hydroxyl and carboxyl ends.

Response:

We performed an end-capping reaction on oligomers with succinic anhydride (carboxyl acid groups) and stopped the reaction when we reached 50% conversion, i.e., the final oligomers had 50:50 COOH:OH functional groups (stoichiometrically balanced). We have now better clarified that in the manuscript (page 5).

..., "followed by end-capping with succinic anhydride (SA) which was stopped at 50% end-group conversion, Fig. 1a and Extended data Table 1."

"The end-capping reaction was monitored by ^1H NMR; after 72h, 50 % end-group conversion of hydroxyl to carboxylic acid was achieved, Extended data Table 1 entries [2] and [3]."

..."multifunctional oligomers are three-armed aliphatic ester oligomers with two types of functional groups: carboxylic acids and hydroxyls (stoichiometrically balanced), Fig. 1d."

Question 6: On page 7, lines 154-155 (page 8, lines 177-178), have the lengths and diameters of WF and MFLC used in this paper been tested? The datas given in this paper were those in the literature.

Response:

Yes, we have previously measured the dimensions of WF and MFLC; the data is now clearly mentioned in SI and in the Methods part (page 19). In addition, we have now better explained the dimensions in the manuscript (pages 8-9).

Page 8: "The wood fibers in this study are 39 μm in diameter and 2.5 mm in length on average²⁴, whereas the present MFLC is 2.5-1000nm in diameter and 5-50 μm in length⁶."

Page 9: "Wood fibers (spruce) are typically 30 μm by 3 mm⁵. The average dimensions of the wood fibers in present study are 39 μm by 2.5 mm. The MFLC fibrils are 2.5 nm to 1 μm thick and several micrometers long. The width of 44wt% of fibrils is less than 100 nm, and the width of 56wt% is between 100 nm and 1 μm ."

Page 19: "The wood fibers had an average length of 2.46 mm and average width of 39.1 μm ^{5,6}. Fiber fine (length < 0.1 mm) content, i.e. length-weighted average was 13.2%. The fibers were beaten and fibrillated into microfibrillated lignocellulose (MFLC) consisting of 44 wt% fibrils with nanosized (<100nm) thickness and rest coarse fibrils with up to 1 μm thickness, as described in ref ⁶. All populations of MFLC fibrils were used."

SI:

Figure S34 Fiber distribution dimensions measured in 20,000 counts using an L&W Fibertester Plus (color shows fiber per mil).

Question 7: "Fig. 1a and b" on page 8, lines 177 may be wrong.

Response:

That is referring to "Extended data Fig. 1a and b". The word "data" was missing, and now is corrected.

Question 8: Is the porosity (density) of the material closely related to the hot pressing pressure? Larger pressure (40-50 MPa) was used in this paper.

Response:

We did not try larger pressures than 40-50 MPa due to instrument limitations. We assume, however, that increasing the pressure will not result in any further reduction in porosity because the polymer's high gel content (cross-linking) is the main limiting factor for melting behavior and consolidation and also this pressure is very high compared to what used for pure c-PCL consolidation and shaping (1 MPa as mentioned in line 488).

Question 9: Is there a comparison of mechanical properties between the hot-pressed wood fiber sheet and c-PCL/HP-WF77?

Response:

The tensile properties of hot-pressed wood fiber sheet and c-PCL/HP-WF77 are compared in SI (Figure S33b). The following explanation is now included in the manuscript (page 13), and in SI (page 23).

“Note that the mechanical properties of c-PCL/HP-WF77 are much lower than those of neat HP-WF given the incorporation of a low-modulus polymer matrix; see SI.”

SI:

“

Figure S33 (b)

The apparent total porosity of HP-WF and c-PCL/HP-WF77 was 23% and 13%, respectively. As a result, the volume fractions of these composites are calculated as: HP-WF (77 v% lignocellulose fiber, 23 v% porosity) and c-PCL/HP-WF77 (67 v% lignocellulose fiber, 20 v% c-PCL, 13 v% porosity). Figure S33b shows that the modulus and tensile strength of c-PCL/HP-WF77 are significantly lower than those of HP-WF (6.6 vs 13.2 GPa for modulus, and 82 vs 154 MPa for strength). This reduction in mechanical properties is understandable given that a soft (low modulus) polymer matrix is used.”

Question 10: Did the degradation test reflect true degradation? What is the ordinate of Figure 4(a)?

Response:

We have used accelerated hydrolytic degradation. We have done this to demonstrate the possibility and potential to chemical recycling. In addition, we have studied relative degradation rate in composites and pure polymer. The comparison of relative degradation rate should be relevant in non-accelerated hydrolytic degradation (in mild conditions). However, biological degradation may have a different result as the mechanism is governed by microorganisms and enzymes.

References

- 1 Sheldon, R. A. The E factor 25 years on: the rise of green chemistry and sustainability. *Green Chemistry* **19**, 18-43, doi:10.1039/C6GC02157C (2017).
- 2 Trost, B. M. The atom economy-a search for synthetic efficiency. *Science* **254**, 1471-1477 (1991).

- 3 Curzons, A. D., Constable, D. J. C., Mortimer, D. N. & Cunningham, V. L. So you think your
process is green, how do you know?—Using principles of sustainability to determine what is
green—a corporate perspective. *Green Chemistry* **3**, 1-6, doi:10.1039/B007871I (2001).
- 4 Oliaei, E., Lindström, T. & Berglund, L. A. Sustainable development of hot-pressed all-
lignocellulose composites—comparing wood fibers and nanofibers. *Polymers* **13**, article
2747, doi:10.3390/polym13162747 (2021).
- 5 Oliaei, E., Berthold, F., Berglund, L. A. & Lindström, T. Eco-friendly high-strength composites
based on hot-pressed lignocellulose microfibrils or fibers. *ACS Sustainable Chemistry &
Engineering* **9**, 1899-1910, doi:10.1021/acssuschemeng.0c08498 (2021).
- 6 Oliaei, E. *et al.* Microfibrillated lignocellulose (MFLC) and nanopaper films from unbleached
kraft softwood pulp. *Cellulose* **27**, 2325-2341, doi:10.1007/s10570-019-02934-8 (2020).
- 7 Panaitescu, D. M., Frone, A. N. & Chiulan, I. Nanostructured biocomposites from aliphatic
polyesters and bacterial cellulose. *Industrial Crops and Products* **93**, 251-266,
doi:10.1016/j.indcrop.2016.02.038 (2016).
- 8 Albertsson, A.-C. & Hakkarainen, M. Designed to degrade. *Science* **358**, 872-873,
doi:doi:10.1126/science.aap8115 (2017).
- 9 Sperling, L. H. *Introduction to Physical Polymer Science*. (John Wiley & Sons, Inc., 2005).
- 10 Figueiredo, A. R. P. *et al.* Antimicrobial bacterial cellulose nanocomposites prepared by in
situ polymerization of 2-aminoethyl methacrylate. *Carbohydrate Polymers* **123**, 443-453,
doi:<https://doi.org/10.1016/j.carbpol.2015.01.063> (2015).
- 11 Yang, X., Berthold, F. & Berglund, L. A. High-density molded cellulose fibers and transparent
biocomposites based on oriented holocellulose. *ACS Applied Materials & Interfaces* **11**,
10310-10319, doi:10.1021/acami.8b22134 (2019).
- 12 Yu, J., Wang, C., Wang, J. & Chu, F. In situ development of self-reinforced cellulose
nanocrystals based thermoplastic elastomers by atom transfer radical polymerization.
Carbohydrate Polymers **141**, 143-150, doi:<https://doi.org/10.1016/j.carbpol.2016.01.006>
(2016).
- 13 Ma, R. *et al.* Application of poly(lactic acid)-grafted cellulose nanofibers as both inhibitor and
reinforcement for 3D-printable tough polydicyclopentadiene composites via frontal ring-
opening metathesis polymerization. *Industrial Crops and Products* **186**, 115217,
doi:<https://doi.org/10.1016/j.indcrop.2022.115217> (2022).
- 14 Carlmark, A., Larsson, E. & Malmström, E. Grafting of cellulose by ring-opening
polymerisation – A review. *European Polymer Journal* **48**, 1646-1659,
doi:10.1016/j.eurpolymj.2012.06.013 (2012).
- 15 Hafrén, J. & Córdova, A. Direct organocatalytic polymerization from cellulose fibers.
Macromolecular Rapid Communications **26**, 82-86, doi:10.1002/marc.200400470 (2005).
- 16 Lönnberg, H. *et al.* Grafting of cellulose fibers with poly(ϵ -caprolactone) and poly(l-lactic
acid) via ring-opening polymerization. *Biomacromolecules* **7**, 2178-2185,
doi:10.1021/bm060178z (2006).
- 17 Lönnberg, H., Fogelström, L., Berglund, L., Malmström, E. & Hult, A. Surface grafting of
microfibrillated cellulose with poly(ϵ -caprolactone) – synthesis and characterization.
European Polymer Journal **44**, 2991-2997, doi:10.1016/j.eurpolymj.2008.06.023 (2008).
- 18 Habibi, Y. *et al.* Bionanocomposites based on poly(ϵ -caprolactone)-grafted cellulose
nanocrystals by ring-opening polymerization. *Journal of Materials Chemistry* **18**, 5002-5010,
doi:10.1039/B809212E (2008).
- 19 Barkoula, N. M., Garkhail, S. K. & Peijs, T. Biodegradable composites based on
flax/polyhydroxybutyrate and its copolymer with hydroxyvalerate. *Industrial Crops and
Products* **31**, 34-42, doi:10.1016/j.indcrop.2009.08.005 (2010).
- 20 Kuciel, S., Mazur, K. & Jakubowska, P. Novel biorenewable composites based on poly (3-
hydroxybutyrate-co-3-hydroxyvalerate) with natural fillers. *Journal of Polymers and the
Environment* **27**, 803-815, doi:10.1007/s10924-019-01392-4 (2019).

- 21 Mármol, G., Gauss, C. & Fanguero, R. Potential of cellulose microfibrils for PHA and PLA biopolymers reinforcement. *Molecules* **25**, article 4653, doi:10.3390/molecules25204653 (2020).
- 22 Terzopoulou, Z. N. *et al.* Green composites prepared from aliphatic polyesters and bast fibers. *Industrial Crops and Products* **68**, 60-79, doi:10.1016/j.indcrop.2014.08.034 (2015).
- 23 Shibata, M., Yosomiya, R., Ohta, N., Sakamoto, A. & Takeishi, H. Poly(ϵ -caprolactone) composites reinforced with short abaca fibres. *Polymers and Polymer Composites* **11**, 359-367, doi:10.1177/096739110301100502 (2003).
- 24 Sjöström, E. V. *Wood Chemistry: Fundamentals and Applications*. (Elsevier Science & Technology, 1993).

Reviewers' Comments:

Reviewer #1:

Remarks to the Author:

The authors have revised the manuscript. As a result, the manuscript is ready for publication. Here are some comments:

- 1) In regard to my comment about green polymerizations and sustainability, here is an observation: the authors state in the letter that "However, for large-scale production, the use of solvents is not needed (at 140 °C the oligomers are liquid and can diffuse to fiber networks)." In the revised manuscript, the authors state "However, for large-scale production, these issues are not so relevant, and E factor is potentially much lower. The use of solvents is not necessary, and non-reacted reagents can potentially be recycled. Therefore, from a large-scale perspective, E factor for the polymerization part is almost zero (near ideal) since everything that remains in the system is desired, and there is no waste or byproducts." I agree that the oligomers are liquid at 140 C. If available, knowing the viscosity of the oligomers would be useful to gauge whether the authors assumption that solvents are not necessary is viable.
- 2) The authors have provided a few green metrics to support their claim of "low environmental impact."
- 3) The revised manuscript and title more accurately reflect the experimental design in Figures 1-2. The rewrite on biodegradation and circular economy combined with the added references are also helpful.
- 4) The accelerated degradation studies are probably the weakest part of the revisions, but marginally satisfactorily.
- 5) Clarification of the CL polymerization with MSA is reasonable.

Reviewer #2:

Remarks to the Author:

The comments of this reviewer have been satisfactorily addressed. The only comment below refers to a piece of information that appears twice and separated by only a few lines in the text. It does not affect the analysis or the discussion and is commented below.

Lines 175-176: "The wood fibers in this study are 39 μm in diameter and 2.5 mm in length on average, whereas the present MFLC is 2.5-1000nm in diameter and 5-50 μm in length"
and

Lines 198-200: "The average dimensions of the wood fibers in the present study are 39 μm by 2.5 mm. The MFLC fibrils are 2.5 nm to 1 μm thick and 200 several micrometers long" contain the same information. Consider to delete one of the two.

Reviewer #3:

Remarks to the Author:

In this paper, lignocellulose biocomposites were prepared by a simple hot pressing method with better mechanical properties. The paper has certain innovation, which may be more suitable for a more professional journal. For all the questions raised, the author has given detailed answers and made revisions. The SI document is also supplemented accordingly. This articlepaper can be published.

Reviewer #1

The authors have revised the manuscript. As a result, the manuscript is ready for publication. Here are some comments:

1) In regard to my comment about green polymerizations and sustainability, here is an observation: the authors state in the letter that "However, for large-scale production, the use of solvents is not needed (at 140 °C the oligomers are liquid and can diffuse to fiber networks)." In the revised manuscript, the authors state "However, for large-scale production, these issues are not so relevant, and E factor is potentially much lower. The use of solvents is not necessary, and non-reacted reagents can potentially be recycled. Therefore, from a large-scale perspective, E factor for the polymerization part is almost zero (near ideal) since everything that remains in the system is desired, and there is no waste or byproducts." I agree that the oligomers are liquid at 140 C. If available, knowing the viscosity of the oligomers would be useful to gauge whether the authors assumption that solvents are not necessary is viable.

Response:

We used a flow sweep measurement to determine the molten oligomer's viscosity at 140 °C.

SI:

Supplementary Figure 37 Viscosity versus shear rate of caprolactone oligomer at 140 °C, measured using a TA instrument rheometer, parallel plate at 0.5 mm gap in flow sweep. The black and red dots represent two sets of measurements, and the line serves as a visual aid.

Additionally, line 384 has been changed to read "The solvents may not be necessary during infiltration as the molten oligomer at 140 °C displays viscosity values similar to commercial epoxy resin formulations (see Supplementary Figure 37)...".

2) The authors have provided a few green metrics to support their claim of "low environmental impact."

Response: Here, we determined several important green chemistry metrics and showed a rather low environmental impact. The unbleached Kraft wood fibers' environmental factors are relatively low. Many other green chemistry metrics take other factors into account. Therefore, we mentioned: "low environmental impact using key green chemistry metrics (atom economy, reaction mass efficiency, and E factor)." For accuracy, we modified low environmental impact to "relatively low environmental impact" in two other places where we discussed eco-indicators, including energy, global warming potential, and water.

3) The revised manuscript and title more accurately reflect the experimental design in Figures 1-2. The rewrite on biodegradation and circular economy combined with the added references are also helpful.

Response:

Thank you.

4) The accelerated degradation studies are probably the weakest part of the revisions, but marginally satisfactorily.

Response:

We agree that accelerated degradation might not be an accurate reflection of the absolute faith of these materials in nature. Nonetheless, after polymerization, these materials are hydrolytically degradable, and the rate of degradation increases in the biocomposite formulation, which can be useful for recycling and circular economy. As a future research direction, it would be interesting to explore if natural fibers could be a pathway for increasing the circularity aspect of materials under industrial composting conditions.

5) Clarification of the CL polymerization with MSA is reasonable.

Response:

Thank you.

Reviewer #2

The comments of this reviewer have been satisfactorily addressed. The only comment below refers to a piece of information that appears twice and separated by only a few lines in the text. It does not affect the analysis or the discussion and is commented below.

Lines 175-176: "The wood fibers in this study are 39 μm in diameter and 2.5 mm in length on average, whereas the present MFLC is 2.5-1000nm in diameter and 5-50 μm in length"

and

Lines 198-200: "The average dimensions of the wood fibers in the present study are 39 μm by 2.5 mm. The MFLC fibrils are 2.5 nm to 1 μm thick and 200 several micrometers long" contain the same information. Consider to delete one of the two.

Response:

The latter sentence has been removed. Thanks.

Reviewer #3

In this paper, lignocellulose biocomposites were prepared by a simple hot pressing method with better mechanical properties. The paper has certain innovation, which may be more suitable for a more professional journal. For all the questions raised, the author has given detailed answers and made revisions. The SI document is also supplemented accordingly. This articlepaper can be published.

Response:

Thank you.